# A COMMUNICATION EFFICIENT FEDERATED KERNEL $k$-MEANS

## ABSTRACT

A federated kernel $k$-means algorithm is developed in this paper. This algorithm resolves two challenging issues: 1) how to distributedly solve the optimization problem of kernel $k$-means under federated settings; 2) how to maintain communication efficiency in the algorithm. To tackle the first challenge, a distributed stochastic proximal gradient descent (DSPGD) algorithm is developed to determine an approximate solution to the optimization problem of kernel $k$-means. To tackle the second challenge, a communication efficient mechanism (CEM) is designed to reduce the communication cost. Besides, the federated kernel $k$-means provides two levels of privacy preservation: 1) users' local data are not exposed to the cloud server; 2) the cloud server cannot recover users' local data from the local computational results via matrix operations. Theoretical analysis shows: 1) DSPGD with CEM converges with an $O(1/T)$ rate, where $T$ is the number of iterations; 2) the communication cost of DSPGD with CEM is unrelated to the number of data samples; 3) the clustering quality of the federated kernel $k$-means approaches that of the standard kernel $k$-means, with a $(1 + \epsilon)$ approximate ratio. The experimental results show that the federated kernel $k$-means achieves the highest clustering quality with the communication cost reduced by more than $60\%$ in most cases.

## 1 INTRODUCTION

Conventionally, kernel $k$-means (Dhillon et al., 2004) is conducted in a centralized manner where training data are stored in one place, such as a cloud server. However, as a rapidly growing number of devices are connected to the Internet, the volume of generated data increase exponentially (Chiang & Zhang, 2016). Uploading all these data to the cloud server can lead to large cost of communication bandwidth. For example, a smartphone manufacturer usually needs to analyze usage patterns of its smartphones, purposing to optimize energy consumption performance of these smartphones. The usage patterns can be obtained by clustering users' energy consumption data via kernel $k$-means. However, if the number of users reaches the order of millions, it may not be a cost-effective scheme to upload all the users' energy consumption data to the cloud server. Besides, uploading users' raw data to the cloud server can lead to data privacy issues. To resolve these issues, a promising approach is to develop a distributed kernel $k$-means algorithm that can be executed under federated settings (McMahan et al., 2017; Yang et al., 2019) where raw data are maintained by users and the cloud has no access to the raw data. In this algorithm, a local training process is conducted at each user's device, based on the local data only. The local computational results, rather than the local data, are then uploaded to the cloud server to accomplish the kernel $k$-means clustering. During this procedure, users' local data are no longer exposed to the cloud server, which provides a basic level of privacy. Besides, it is usually more communication efficient to upload the local computational results than to upload the local data to the cloud server.

However, it is nontrivial to design a federated learning algorithm for kernel $k$-means due to three challenging issues: 1) how to solve the optimization problem of kernel $k$-means in a distributed manner without sending users' data to a central place; 2) how to maintain communication efficiency in the algorithm; 3) how to protect users' data privacy in the algorithm. Considering the first issue under federated settings, the key problem is to obtain the top eigenpairs of the kernel matrix $\mathbf{K}$ (as required by kernel $k$-means) in a distributed manner. To solve this problem, a distributed stochastic proximal gradient descent (DSPGD) algorithm is developed as follows. Since $\mathbf{K}$ is not available

under federated settings, an estimate of $\mathbf{K}$, denoted as $\boldsymbol{\xi}$, is first constructed distributively at users' devices based on random features (Rahimi & Recht, 2008) of local data samples. Since the estimate is distributed among different devices, it is processed by the distributed Lanczos algorithm (DLA) (Penna & Stańczak, 2014) to obtain an estimate of $\mathbf{K}$ (denoted as $\mathbf{Z}$) at the cloud server. Afterwards, an approximate version of the top eigenpairs of $\mathbf{K}$ can be obtained from $\mathbf{Z}$ through singular value decomposition (SVD). To improve the accuracy of approximation, the former steps are conducted in an iterative way. More specifically, in the $t$-th iteration, an estimate $\boldsymbol{\xi}_t$ is constructed at users' devices, and then the estimate $\mathbf{Z}_t$ at the cloud server is updated to $\mathbf{Z}_{t+1}$ via stochastic proximal gradient descent (SPGD) (Zhang et al., 2016). It is proved that, after sufficient iterations, $\mathbf{Z}_t$ can converge to a low rank matrix whose top eigenpairs are the same as those of $\mathbf{K}$[1] As a result, top eigenpairs of $\mathbf{K}$ are finally obtained at the cloud server.

To resolve the second issue, DLA operations in DSPGD need to be enhanced to reduce communication cost. When DLA is executed in DSPGD, the process of obtaining an updated $\mathbf{Z}_t$ at the cloud server results in high communication cost between users' devices and the cloud server, because the operation is conducted upon matrices (e.g., $\boldsymbol{\xi}_t$) with the number of rows/columns equal to the number of data samples. To prevent the communication cost from growing with the number of data samples, a communication efficient mechanism (CEM) is designed so that DLA is operated upon a different type of matrices whose dimensions are reduced and independent of the number of data samples. More specifically, a new matrix $\mathbf{W}_t$ is designed such that: 1) $\mathbf{W}_t\mathbf{W}_t^\top$ has the same eigenvectors as those of $\mathbf{Z}_{t+1}$, but its eigenvalues are smaller by a constant; 2) $\mathbf{W}_t$ and $\mathbf{Z}_t$ can be constructed distributively at users' devices based on local values of $\boldsymbol{\xi}_t$. Furthermore, DLA is applied to $\mathbf{W}_t^\top\mathbf{W}_t$ (instead of $\mathbf{W}_t\mathbf{W}_t^\top$), so its operations are performed upon matrices with a highly reduced dimension. Via DLA operations between users' devices and the cloud server, $\mathbf{W}_t$ and $\mathbf{Z}_t$ are updated iteratively, and then the top eigenpairs of $\mathbf{W}_t^\top\mathbf{W}_t$ are obtained at users' devices. Once $\mathbf{Z}_t$ converges, users' devices transform the top eigenpairs of $\mathbf{W}_t^\top\mathbf{W}_t$ to those of $\mathbf{W}_t\mathbf{W}_t^\top$ and further obtain the eigenpairs of $\mathbf{Z}_t$. Instead of sending these eigenpairs to the cloud server, a distributed linear k-means algorithm (Balcan et al., 2013) is incorporated into CEM so that the cloud server can perform clustering directly on the eigenpairs of the converged $\mathbf{Z}_t$. As shown in the process of CEM, the communication efficiency of DSPGD is significantly improved.

For the third issue, FK $k$-means based on DSPGD and CEM provides two levels of privacy preservation: 1) users' local data are not exposed to the cloud server; 2) the cloud server cannot recover users' local data from the local computational results via matrix operations. To provide stronger privacy, a differential privacy mechanism (Dwork et al., 2006) needs to be integrated with FK $k$-means, which is subject to future study.

The theoretical analysis shows that DSPGD with CEM converges to $\mathbf{Z}^*$ at an $O(1/T)$ rate, where $T$ is the iteration number. The communication cost of DSPGD with CEM is linear to the dimension of the right singular vector times the number of users, which can be much smaller than the number of data samples. The clustering quality of the federated kernel $k$-means approaches that of kernel $k$-means, with a $(1+\epsilon)$ approximate ratio. The experimental results show that, compared with the state-of-the-art schemes, FK $k$-means achieves the highest clustering quality with the communication cost reduced by more than $60\%$ in most cases.

## 2 RELATED WORK

### 2.1 DISTRIBUTED KERNEL $k$-MEANS

Many algorithms have been developed to conduct the kernel $k$-means clustering in a distributed way. The kernel approximation method is a popular approach employed in these algorithms, such as the Nyström method (Chitta et al., 2011; 2014; Wang et al., 2019) and the random feature method (Chitta et al., 2012). A trimmed kernel $k$-means algorithm (Tsapanos et al., 2015) decreases the computational cost and the space complexity by significantly reducing the non-zero entries in $K$ via a kernel matrix trimming algorithm. In (Elgohary et al., 2014) an approximate nearest centroid (APNC) embedding is developed to embed the data samples so that the clustering assignment step

---

[1]More specifically, the top eigenvectors of the low rank matrix are the same as those of $\mathbf{K}$ and its nonzero eigenvalues are smaller than those of $\mathbf{K}$ by a constant.

of kernel $k$-means can be parallel executed. A communication efficient kernel principle component analysis (PCA) algorithm (Balcan et al., 2016) along with distributed liner $k$-means can approximately solve the optimization problem of kernel $k$-means while maintaining the communication efficiency. However, these algorithms are designed with an assumption that they are executed at the cloud server where users' raw data are collected. Besides, many of these algorithms (Chitta et al., 2011; 2012; 2014; Wang et al., 2019) are one-shot algorithms, i.e., they only determine an approximate kernel matrix $\tilde{\mathbf{K}}$ once. Thus, their clustering quality is limited by the accuracy of $\tilde{\mathbf{K}}$. In contrast to these algorithms, FK $k$-means is the first distributed kernel $k$-means scheme designed under federated settings. In addition, FK $k$-means is an iterative algorithm that can approach the top eigenpairs of $\mathbf{K}$ more accurately by employing more iterations.

## 2.2 FEDERATED LEARNING

Federated learning (McMahan et al., 2017) is a new machine learning framework aiming to protect users' data privacy and save the communication cost during the learning process. In the framework, a local model is updated at each user's device, and these local models instead of users' local data are then aggregated at the cloud server to generate a global model. The distributed optimization method in the framework is applicable to the models whose optimization problem can be decomposed into several independent subproblems, such as neural networks. and many algorithms (Konečný et al., 2016; Yang et al., 2018; Yurochkin et al., 2019) are developed. However, it is non-trivial to decompose the optimization problem of kernel $k$-means under the federated learning framework. Some algorithms (Liu et al., 2017; Caldas et al., 2018) improve federated multi-task learning (Smith et al., 2017) with kernel. However, these algorithms either employ explicit feature mapping (Liu et al., 2017) that can lead to impractical computational cost, or require to send the support vectors of users' local data (i.e., some local data samples) to the cloud server (Caldas et al., 2018), which can leak users' privacy information. Due to these limitations, these algorithms are not applicable to kernel $k$-means under the federated learning framework. Recently, a concept of clustered federated learning (Ghosh et al., 2020; Sattler et al., 2020; Mansour et al., 2020) is proposed, where the clients are clustered according to their gradient updates or their local models. However, their clustering problems are different from the optimization problem of kernel $k$-means, and thus are not feasible for the optimization problem of kernel $k$-means in the federated setting.

## 2.3 STOCHASTIC KERNEL PCA

In (Zhang et al., 2016), the stochastic kernel PCA is accomplished a stochastic proximal gradient descent (SPGD) algorithm. As a result, SPGD is a centralized counterpart of the distributed proximal gradient descent (DSPGD) algorithm in FK $k$-means. However, DSPGD is distinct from SPGD in three features. First, DSPGD is conducted under federated settings while SPGD is conducted in a centralized manner where users' raw data are collected at the cloud server. Second, the communication cost is considered in the design of DSPGD, which results in CEM, while the communication cost is not considered in SPGD. Third, although both DSPGD and the SPGD aim at approaching the top eigenpairs of $\mathbf{K}$, in the $t$-th iteration, DSPGD only needs to obtain an approximate solution $\bar{\mathbf{Z}}_{t+1}$ to the problem of updating $\mathbf{Z}_t$ instead of the exact solution $\mathbf{Z}_{t+1}^*$ to the same problem like SPGD, which leads to less communication cost under federated settings.

## 3 PRELIMINARY

Let $\{\mathbf{x}_i\}_{i=1}^N \subseteq \mathcal{X}$ be a set of $N$ data samples. Given a feature mapping $\phi(\cdot) : \mathcal{X} \mapsto \mathcal{H}$ and the number of clusters $k$, the problem of kernel $k$-means whose objective is to find an optimal indicator matrix $\mathbf{Y}^*$ can be written as

$$\min_{\mathbf{Y} \in \{0,1\}^{N \times k}} \mathrm{Tr}(\mathbf{K}) - \mathrm{Tr}(\mathbf{L}^{\frac{1}{2}} \mathbf{Y}^\top \mathbf{K} \mathbf{Y} \mathbf{L}^{\frac{1}{2}}) \quad \text{s.t. } \mathbf{Y}\mathbf{1}_k = \mathbf{1}_n, \tag{1}$$

where $\mathbf{K}$ is the kernel matrix with each entry $K_{ij} = \phi(\mathbf{x}_i)^\top \phi(\mathbf{x}_j)$, $\mathbf{L}^{\frac{1}{2}} = \mathbf{Diag}([\frac{1}{\sqrt{N_1}}, \ldots, \frac{1}{\sqrt{N_k}}])$ is a diagonal matrix, $N_i$ is the number of samples in the $i$-th cluster, and $\mathbf{1}_k$ is a column vector with all the $k$ items equal to 1. However, the problem in equation (1) is an NP-hard problem (Garey et al., 1982; Wang et al., 2019). To this end, an approximate solution $\tilde{\mathbf{Y}}$ is required. An efficient approach

to obtaining the approximate solution is as follows. $\mathbf{K}$ is decomposed as $\mathbf{K} = \mathbf{U}\mathbf{\Lambda}\mathbf{U}^\top$ via eigenvalue decomposition (EVD), and then linear $k$-means is applied to the matrix $\mathbf{H} = \mathbf{U}\mathbf{\Lambda}^{\frac{1}{2}}$ to obtain $\tilde{\mathbf{Y}}$ (Ding et al., 2005; Chitta et al., 2012; Wang et al., 2019). To reduce the computational complexity, only the first $s$ column vectors of $\mathbf{H}$ are selected as the input of linear $k$-means (Cohen et al., 2015).

## 4 FEDERATED KERNEL $k$-MEANS

In FK $k$-means, the approximate solution to the problem in equation (1) is also determined based on the top-$s$ eigenpairs of the kernel matrix $\mathbf{K}$. To obtain these eigenpairs under federated settings, a distributed stochastic proximal gradient descent (DSPGD) algorithm is developed in Section 4.1. A communication efficient mechanism is then designed to reduce the communication cost of DSPGD in Section 4.2.

### 4.1 DISTRIBUTED STOCHASTIC PROXIMAL GRADIENT DESCENT

The key problem for designing FK $k$-means is to obtain the the top-$s$ eigenpairs of $\mathbf{K}$ in a distributed manner. To solve this problem, a distributed stochastic proximal gradient descent algorithm is developed as follows. Under the federated settings, the main challenge on determining the top-$s$ eigenpairs of $\mathbf{K}$ is that $\mathbf{K}$ is not available since users' local data cannot be exposed to the cloud server or other users. To this end, an estimate of $\mathbf{K}$, denoted as $\boldsymbol{\xi}$, is constructed distributively at users' devices based on the random features (Rahimi & Recht, 2008; Kar & Karnick, 2012) of their local data samples. More specifically, $\boldsymbol{\xi} = \frac{1}{D}\mathbf{A}\mathbf{A}^\top$ and $\mathbb{E}[\boldsymbol{\xi}] = \mathbf{K}$, where $D$ is the number of random features of each data samples, and $\mathbf{A} = [\mathbf{A}[1]^\top, \dots, \mathbf{A}[M]^\top]^\top$ is the random feature matrix distributed over $M$ users' devices (the details of the random feature method are included in Appendix A). Since $\boldsymbol{\xi}$ is distributed over users' devices, it is then processed by the distributed Lanczos algorithm (DLA) (Penna & Stańczak, 2014) to obtain an estimate of $\mathbf{K}$, i.e., $\mathbf{Z}$ at the cloud server. Afterwards, an approximate version of the top eigenpairs of $\mathbf{K}$ can be obtained from $\mathbf{Z}$ through SVD. To improve the accuracy of approximation, one method is to increase the value of $D$. However, this method is only feasible when each user's device has enough memory space. To adapt DSPGD to devices with different memory space, DSPGD improves the accuracy of approximation via an iterative method. More specifically, in the $t$-th iteration, an estimate $\boldsymbol{\xi}_t$ is constructed at users' devices, and then the estimate $\mathbf{Z}_t$ at the cloud server is updated to $\mathbf{Z}_{t+1}$ via stochastic proximal gradient descent (Zhang et al., 2016):

$$\mathbf{Z}_{t+1} = \underset{\mathbf{Z}\in\mathbb{R}^{n\times n}}{\arg\min} \frac{1}{2}||\mathbf{Z} - \mathbf{Z}_t||_F^2 + \eta_t\langle\mathbf{Z} - \mathbf{Z}_t, \mathbf{Z}_t - \boldsymbol{\xi}_t\rangle + \eta_t\lambda||\mathbf{Z}||_*,$$

where $\eta_t$ is a learning rate. $\mathbf{Z}_{t+1}$ has an explicit expression $\mathbf{Z}_{t+1} = \sum_{i:\tilde{\lambda}_{i,t}>\eta_t\lambda}(\tilde{\lambda}_{i,t} - \eta_t\lambda)\tilde{\mathbf{u}}_{i,t}\tilde{\mathbf{u}}_{i,t}^\top$, where $(\tilde{\mathbf{u}}_{i,t}, \tilde{\lambda}_{i,t})$ is the $i$-th eigenpair of a matrix $\mathbf{R}_t = (1 - \eta_t)\mathbf{Z}_t + \eta_t\boldsymbol{\xi}_t$. Since $\boldsymbol{\xi}$ is distributed over users' devices, $\mathbf{Z}_{t+1}$ is determined via DLA at the cloud server. It is proved that, after sufficient iterations, $\mathbf{Z}_t$ can converge to a low rank matrix $\hat{\mathbf{K}} = \sum_{i:\lambda_i>\lambda}(\lambda_i - \lambda)\mathbf{u}_i\mathbf{u}_i^\top$ where $(\mathbf{u}_i, \lambda_i)$ is the $i$-th eigenpair of $\mathbf{K}$. As a result, the top eigenpairs of $\mathbf{K}$ are finally obtained at the cloud server.

The $t$-th iteration of DSPGD is executed as follows. The main task is to approach the top eigenpairs of $\mathbf{R}_t$ via DLA. The cloud server first initializes a random vector $\mathbf{c}_1 \in \mathbb{R}^N$. In the $q$-th iteration of DLA, the cloud server determines a vector $\mathbf{g}_q = \mathbf{R}_t\mathbf{c}_q = (1 - \eta_t)\mathbf{Z}_t\mathbf{c}_q + \eta_t\mathbf{A}_t\mathbf{A}_t^\top\mathbf{c}_q/D$, where $\mathbf{Z}_t\mathbf{c}_q$ is computed at the cloud server, and $\mathbf{A}_t\mathbf{A}_t^\top\mathbf{c}_q$ is computed in a distributed manner. The computation of $\mathbf{A}_t\mathbf{A}_t^\top\mathbf{c}_q$ is accomplished by five steps: 1) the cloud server partitions the vector $\mathbf{c}_q = [\mathbf{c}_q[1]^\top, \dots, \mathbf{c}_q[M]^\top]^\top$ into $M$ parts and sends the $m$-th part $\mathbf{c}_q[m]$ to the user $m$; 2) the user $m$ computes a local vector $\mathbf{A}_t[m]^\top\mathbf{c}_q[m]$ and uploads the vector to the cloud server; 3) the cloud server sums up these vectors to obtain the vector $\mathbf{A}_t^\top\mathbf{c}_q$ and then broadcasts this vector to all the users; 4) the $m$-th user computes a new local vector $\mathbf{A}_t[m]\mathbf{A}_t^\top\mathbf{c}_q$ and then uploads this vectors to the cloud server; 5) the cloud server finally concatenates these vectors from users to form $\mathbf{A}_t\mathbf{A}_t^\top\mathbf{c}_q$; When $\mathbf{g}_q$ is determined, the cloud server then applies the Lanczos algorithm to the collected vectors $\{\mathbf{g}_1, \dots, \mathbf{g}_q\}$ to approximate the top eigenpairs of $\mathbf{R}_t$ (the details about the Lanczos algorithm and

the complete procedure of DLA are provided in Appendix B). After sufficient iterations of DLA, the top eigenpairs of $\mathbf{R}_t$ are obtained at the cloud server, and then $\mathbf{Z}_{t+1}$ is determined accordingly.

## 4.2 COMMUNICATION EFFICIENT MECHANISM

When DLA is executed in DSPGD, the process of obtaining an updated $\mathbf{Z}_t$ at the cloud server results in high communication cost between the cloud server and users' devices, because its operation is upon matrices (e.g., $\boldsymbol{\xi}_t$) with the number of rows/columns equal to the number of data samples. To prevent the communication cost from growing with the number of data samples, a communication efficient mechanism (CEM) is designed so that DLA is operated upon a different type of matrices whose dimensions are independent from the number of data samples.

A new matrix $\mathbf{W}_t$ that satisfies $\mathbf{W}_t \mathbf{W}_t^\top$ equals $\mathbf{R}_t$ is designed as follows. Let $\mathbf{Z}_t = \tilde{\mathbf{U}}_t \tilde{\boldsymbol{\Lambda}}_t \tilde{\mathbf{U}}_t^\top$ be the eigendecomposition of $\mathbf{Z}_t$, and $\mathbf{B}_t$ equal $\tilde{\mathbf{U}}_t \tilde{\boldsymbol{\Lambda}}_t^{\frac{1}{2}}$. Based on $\mathbf{B}_t$ and the random feature matrix $\mathbf{A}_t$, $\mathbf{W}_t$ is constructed as $\mathbf{W}_t = [\sqrt{\frac{\eta_t}{D}} \mathbf{A}_t, \sqrt{1 - \eta_t} \mathbf{B}_t]$. Assume that $\mathbf{B}_t$ is divided like $\mathbf{A}_t$, i.e., $\mathbf{B}_t = [\mathbf{B}_t[1]^\top, \ldots, \mathbf{B}_t[M]^\top]^\top$, and $\mathbf{B}_t[m]$ is maintained at the $m$-th user's device. As a result, a submatrix of $\mathbf{W}_t$ can be constructed at the $m$-th device by $\mathbf{W}_t[m] = [\sqrt{\frac{\eta_t}{D}} \mathbf{A}_t[m], \sqrt{1 - \eta_t} \mathbf{B}_t[m]]$.

Furthermore, DLA is applied to $\mathbf{W}_t^\top \mathbf{W}_t$ instead of $\mathbf{W}_t \mathbf{W}_t^\top$. The number of rows/columns of $\mathbf{W}_t^\top \mathbf{W}_t$ equals $r_t + D$ where $r_t$ is the rank of $\mathbf{Z}_t$ and $D$ is the number of random features. Compared with the number of data samples $N$, $r_t + D$ is usually much smaller than $N$, so the operation of DLA is upon matrices with a highly reduced dimension. In the $q$-th iteration of DLA, the computation of $\mathbf{g}_q = \mathbf{W}_t^\top \mathbf{W}_t \mathbf{c}_q$ is accomplished by three steps: 1) the cloud server first broadcasts a vector $\mathbf{c}_q$; 2) each user $m$ computes a local vector $\mathbf{W}_t[m]^\top \mathbf{W}_t[m] \mathbf{c}_q$ and uploads the vector to the cloud server; 3) the cloud server sums up these vectors to obtain $\mathbf{g}_q$. The cloud server then transforms $\mathbf{g}_q$ into $\mathbf{c}_{q+1}$ following the Lanczos iteration. After sufficient iterations of DLA, the top-$s_t$ eigenpairs $\{(\tilde{\lambda}_{i,t}, \tilde{\mathbf{v}}_{i,t}), i = 1, \ldots, s_t\}$ of $\mathbf{W}_t^\top \mathbf{W}_t$ converge at the cloud server. The cloud server then broadcasts the obtained top-$s_t$ eigenpairs to all the users' devices. Since the user $m$ keeps $\mathbf{W}_t[m]$, it then determines $s_t$ vectors $\tilde{\mathbf{u}}_{i,t}[m] = \frac{1}{\sqrt{\tilde{\lambda}_{i,t}}} \mathbf{W}_t[m] \tilde{\mathbf{v}}_{i,t}, i = 1, \ldots, s_t$, where $\tilde{\mathbf{u}}_{i,t}[m]$ is one part of the eigenvector $\tilde{\mathbf{u}}_{i,t}$, i.e., $\tilde{\mathbf{u}}_{i,t} = [\tilde{\mathbf{u}}_{i,t}[1]^\top, \ldots, \tilde{\mathbf{u}}_{i,t}[M]^\top]^\top$. Based on the vectors $\{\tilde{\mathbf{u}}_{i,t}[m], i = 1, \ldots, s_t\}$ and the eigenvalues $\{\tilde{\lambda}_{i,t}, i = 1, \ldots, s_t\}$, the user $m$ also determines a submatrix $\mathbf{B}_{t+1}[m]$ of $\mathbf{B}_{t+1}$ via $\mathbf{B}_{t+1}[m] = [\sqrt{\tilde{\lambda}_{1,t} - \eta_t \lambda} \tilde{\mathbf{u}}_{1,t}[m], \ldots, \sqrt{\tilde{\lambda}_{i-1,t} - \eta_t \lambda} \tilde{\mathbf{u}}_{i-1,t}[m]]$, which enables the $(t+1)$-th iteration of DSPGD with CEM.

As DSPGD with CEM converges after $T$ iterations, a matrix $\mathbf{H}[m]$ is constructed at the $m$-th user's device by $\mathbf{H}[m] = [\sqrt{\tilde{\lambda}_{1,T} + (1 - \eta_T)\lambda} \tilde{\mathbf{u}}_{1,T}[m], \ldots, \sqrt{\tilde{\lambda}_{s,T} + (1 - \eta_T)\lambda} \tilde{\mathbf{u}}_{s,T}[m]]$. A distributed linear $k$-means algorithm (Balcan et al., 2013) is then applied to the rows of $\mathbf{H} = [\mathbf{H}[1]^\top, \ldots, \mathbf{H}[M]^\top]^\top$ to obtain the clustering result. DSPGD with CEM and the distributed linear $k$-means algorithm constitute FK $k$-means. The pseudo code of FK $k$-means is shown in Algorithm 1.

## 5 THEORETICAL ANALYSIS

The convergence of DSPGD with CEM is analyzed in Section 5.1. The communication cost of CEM is analyzed in Section 5.2, which shows CEM is important for FK $k$-means to maintain the communication efficiency. It is then proved that the clustering quality of FK $k$-means can approach that of the standard kernel $k$-means in Section 5.3. Besides, the privacy preservation provided by FK $k$-means is analyzed in Section 5.4.

### 5.1 CONVERGENCE ANALYSIS FOR DSPDG

The convergence rate of DSPGD with CEM is derived in Theorem 1.

**Theorem 1.** *Define* $\gamma = \max_{t \in [T]} ||\mathbf{Z}_t||_*$ *and* $C^2 = \max_{t \in [T]} ||\mathbf{Z}_t - \boldsymbol{\xi}_t||_F^2$. *Assume* $||\boldsymbol{\xi}_t - \mathbf{K}||_F \leq G$, *and* $||\mathbf{Z}_t - \mathbf{Z}^*||_F \leq H, \forall t > 2$. *By setting* $\eta_t = 2/t$, *the following upper bound of*

---

**Algorithm 1** Federated Kernel $k$-Means Algorithm

---

1: **Input:** The threshold parameter $\lambda$, the number of eigenvectors $s$ to be approached, the number of random features $D$, the maximal number of iterations $T$, local datasets $\mathcal{L}_m, m = 1, ..., M$, the initial local matrix $\mathbf{B}_1[m] = \mathbf{0}, m = 1, ..., M$
2: **Output:** the clustering assignment for each data sample
3: **Server executes:**
4: **for** $t = 1, 2, \ldots, T$ **do**
5:     Initialize $\eta_t = 1/t$, $q = 0$
6:     **for** each client $m$, $m = 1, 2, ..., M$ **in parallel do**
7:         Compute $\mathbf{A}_t[m]$ by applying a random feature method to $\mathcal{L}_m$
8:         Construct $\mathbf{W}_t[m] = [\sqrt{\frac{\eta_t}{D}}\mathbf{A}_t[m], \sqrt{1-\eta_t}\mathbf{B}_t[m]]$
9:     **end for**
10:    Call DLA to determine the eigenpairs $\{(\tilde{\lambda}_{i,t}, \tilde{\mathbf{v}}_{1,t}), i = 1, ..., s_t\}$ of $\mathbf{W}_t^\top \mathbf{W}_t$
11:    **for** each client $m$, $m = 1, 2, ..., M$ **in parallel do**
12:        Compute $\tilde{\mathbf{u}}_{i,t}[m] = \frac{1}{\sqrt{\tilde{\lambda}_{i,t}}}\mathbf{W}_t[m]\tilde{\mathbf{v}}_{i,t}$ for $i = 1, ..., s_t$
13:        Compute $\mathbf{B}_{t+1}[m] = [\sqrt{\tilde{\lambda}_{1,t} - \eta_t\lambda}\tilde{\mathbf{u}}_{1,t}[m], ..., \sqrt{\tilde{\lambda}_{s_t,t} - \eta_t\lambda}\tilde{\mathbf{u}}_{s_t,t}[m]]$
14:    **end for**
15: **end for**
16: **for** each client $m$, $m = 1, 2, ..., M$ **in parallel do**
17:    Construct $\mathbf{H}[m] = [\sqrt{\tilde{\lambda}_{1,T} + (1-\eta_T)\lambda}\tilde{\mathbf{u}}_{1,T}[m], ..., \sqrt{\tilde{\lambda}_{s,T} + (1-\eta_T)\lambda}\tilde{\mathbf{u}}_{s,T}[m]]$
18: **end for**
19: Apply a distributed linear $k$-means algorithm over the rows of $\mathbf{H} = [\mathbf{H}[1]^\top, ..., \mathbf{H}[M]^\top]^\top$
20: **Return** clustering assignment for each data sample

---

$||\mathbf{Z}_{T+1} - \mathbf{Z}^*||_F^2$ *holds with a probability at least* $1 - \delta$

$$||\mathbf{Z}_{T+1} - \mathbf{Z}^*||_F^2 \leq \frac{4}{T}\left(C^2 + \lambda\gamma + 2G^2\tau + \frac{2}{3}GH\tau + GH\right) = O(1/T), \qquad (2)$$

*where* $\tau = \log\frac{\lceil 2\log_2 T\rceil}{\delta}$.

The result in Theorem 1 indicates that DSPGD with CEM converges to $\mathbf{Z}^*$ at an $O(1/T)$ rate. The proof of Theorem 1 is provided in Appendix C.

## 5.2 COMMUNICATION COST ANALYSIS FOR CEM

Define the communication cost as the number of floating-point numbers uploaded from users' devices to the cloud server. In Theorem 2, the communication cost of DSPGD with CEM and the communication cost of DSPGD without CEM are both analyzed.

**Theorem 2.** *For DSPGD with CEM, in the $t$-th iteration, its communication cost is linear to $r_t + D$ where $r_t$ is the rank of $\mathbf{Z}_t$ and $D$ is the number of random features. Define the communication ratio as the ratio of the communication cost of DSPGD without CEM to that of DSPGD with CEM. In the $t$-th iteration, the communication ratio equals $\frac{(N+MD)Q_0}{M(r_t+D)Q_1}$ where $N$ is the number of data samples; $M$ is the number of users; $Q_0$ and $Q_1$ are the number of Lanczos iterations for DSPGD without CEM and DSPGD with CEM, respectively.*

By Theorem 2, the communication cost of DSPGD with CEM is unrelated to the number of data samples $N$, and the communication cost reduced by CEM can be revealed by the ratio. The values of $Q_0$ and $Q_1$ are affected by the selection of initial vector $\mathbf{c}_1$. However, empirically the values of $Q_0$ and $Q_1$ are at the same order no matter which initial vectors are chosen. The dominant factor of the ratio is still $\frac{N+MD}{M(r_t+D)}$. Since $\mathbf{Z}_t$ is used to approach the top-$s$ eigenpairs of $\mathbf{K}$, empirically its rank $r_t$ has an upper bound. In our experiments, the value of $r_t$ is at the same order of $s$, i.e. the number of eigenvectors of $\mathbf{K}$ to be determined by DSPGD. Usually, the number of data samples at a user's device is much larger than $s$ so that it is easy to satisfied that $N > Mr_t$, and CEM can definitely reduce the communication cost for DSPGD in these cases. The proof of Theorem 2 and the empirical results for $r_t$ are given in Appendix D.

### 5.3 APPROXIMATE RATIO ANALYSIS FOR FEDERATED KERNEL $k$-MEANS

Before the analysis, a $\gamma$-approximate algorithm is first defined as follows.

**Definition 1.** *A linear $k$-means algorithm is applied to a matrix $\mathbf{H}$ with $n$ row, where an indicator matrix $\tilde{\mathbf{Y}}$ is obtained. This algorithm is called a $\gamma$-approximate algorithm if, for any matrix $\mathbf{H}$, $f(\tilde{\mathbf{Y}}; \mathbf{H}) \leq \gamma \min_{\mathbf{Y}} f(\mathbf{Y}; \mathbf{H})$ where $f$ is the objective function of linear $k$-means.*

It has been proved that the standard kernel $k$-means algorithm (Dhillon et al., 2004) is a $\gamma$-approximate algorithm (Wang et al., 2019). The approximate ratio is then derived in Theorem 3 for FK $k$-means.

**Theorem 3.** *The objective function of kernel $k$-means in (1) is denoted as $f_K$. $\mathbf{H}_T$ is the output of DSPGD with CEM after $T$ iterations, and a $\gamma$-approximate algorithm is applied to the first $s$ columns of $\mathbf{H}_T$ to obtain $\widetilde{\mathbf{Y}}_T$. Assume the assumptions in Theorem 1 hold. For $\widetilde{\mathbf{Y}}_T$, the following inequality holds with a probability at least $1 - \delta(T)$, i.e., $f_K(\widetilde{\mathbf{Y}}_T) \leq \gamma(1 + \varepsilon + \frac{k}{s}) \min_{\mathbf{Y}} f_K(\mathbf{Y})$, where $\varepsilon = O(\sqrt{\frac{s}{T}})$.*

Note that as $T$ increases, $\delta(T)$ decreases. If $s = O(k/\varepsilon)$, $T = O(k/\varepsilon^3)$, then the clustering quality (in terms of the loss $f_K(\mathbf{Y})$) of FK $k$-means approaches that of the standard kernel $k$-means with a $(1 + \varepsilon)$-approximate ratio. The proof of Theorem 3 is given in Appendix E.

### 5.4 PRIVACY ANALYSIS FOR FEDERATED KERNEL $k$-MEANS

FK $k$-means can provide two levels of privacy preservation: 1) one user's local data are not exposed to the cloud server and other users; 2) the cloud server cannot recover users' local data from the collected local computational results via matrix operation. The first level can be easily verified from the procedure of FK $k$-means. The second level is proved by Theorem 4.

**Theorem 4.** *Based on the collected local computational results, the cloud server can at most recover the matrices $\{\mathbf{W}_t[m]^\top \mathbf{W}_t[m], m = 1, ..., M\}$ via matrix operations. Moreover, recovering the random feature matrix $\mathbf{A}_t$ from such matrices is an ill-posed problem with infinite solutions.*

By Theorem 4, the cloud server cannot recover the random feature matrices from the local computational results. Without such random feature matrices, it is infeasible for the cloud server to recover users' local data via matrix operations. More explanation and the proof of Theorem 4 are provided in Appendix F. Moreover, FK $k$-means can incorporate the differential privacy mechanism (Dwork et al., 2006; Su et al., 2016) or random perturbation (Lin, 2016) to provide higher level of privacy preservation, which is subject to future work.

## 6 EXPERIMENTS

### 6.1 EXPERIMENTAL SETTING

Four types of existing schemes are considered in the experiments: centralized kernel $k$-means (Zha et al., 2001) (denoted as CK $k$-means), scalable kernel $k$-means (Wang et al., 2019) (denoted as SK $k$-means), distributed kernel $k$-means with random feature (Chitta et al., 2012) (denoted as RFK $k$-means), and communication efficient distributed kernel PCA (Balcan et al., 2016) (denoted as CE PCA). CK $k$-means and SK $k$-means are executed at the cloud server (denoted as cloud-based algorithms), and the rest methods are executed in a distributed manner where users' raw data cannot be uploaded to the cloud server (denoted as client-based algorithms). Besides, Gaussian kernel is used in each algorithm. Four datasets are selected for performance evaluation: Three public datasets (Mushrooms, MNIST, and Covtype) from the LIBSVM dataset repository and one dataset (Smartphone) provided by a company. In addition, $20,000$ data samples are randomly selected from the dataset MNIST to construct a dataset MNIST-small that is used to validate the convergence of DSPGD. The statistical information of these datasets is given in Table 1. The description of the Smartphone dataset and the existing methods are included in Appendix G.

The hyperparameters of FK $k$-means are determined as follows. The kernel parameter $\gamma$ is computed based on the average interpoint distance in the given dataset (Wang et al., 2019): $\gamma =$

Table 1: Datasets statistics and hyperparameter settings for FK $k$-means

| Dataset | #samples ($N$) | #features ($d$) | #clusters ($k$) | #random features ($D$) | #eigenvectors ($s$) |
|---|---|---|---|---|---|
| Mushrooms | $8,124$ | 112 | 2 | 15 | 2 |
| MNIST-small | $20,000$ | 780 | 10 | 200 | 10 |
| MNIST | $60,000$ | 780 | 10 | 200 | 12 |
| Covtype | $581,012$ | 54 | 7 | 30 | 10 |
| Smartphone | $177,029$ | 12 | 4 | 20 | 6 |

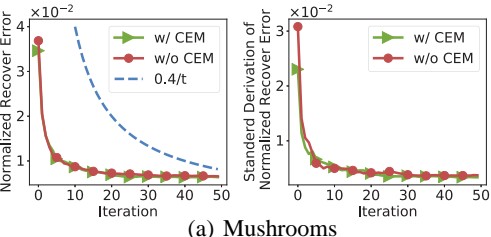

(a) Mushrooms

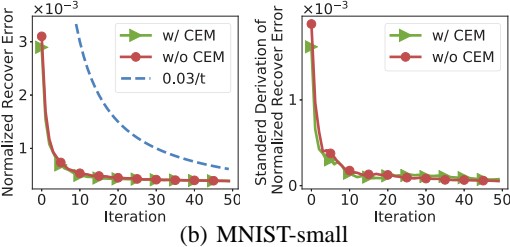

(b) MNIST-small

Figure 1: The convergence curves of the two versions of DSPGD and standard deviation curves of the normalized recover error on the Mushrooms dataset and the MNIST-small dataset.

$\frac{N^2}{2\sum_{i=1}^{N}\sum_{j=1}^{N}\|\mathbf{x}_i-\mathbf{x}_j\|_2^2}$. The threshold parameter $\lambda$ of DSPGD is set to the $(k+2)$-th eigenvalue obtained in the first iteration of DSPGD. The configuration of the number of random features $D$ and the parameter $s$ in the top-$s$ eigenvectors for each dataset is provided in Table 1 (the hyperparameter configuration of the existing methods and a discussion on the configuration of $D$ are provided in Appendix G). In the experiments, $M = 5$ worker processes and one coordinator process are generated to simulate users' devices and the cloud server, respectively. The worker processes communicate with the coordinator process via the message passing interface (MPI) in a synchronized manner. All the experiments are executed in a server with one i7-6850k CPU and 32 GB RAM.

## 6.2 EXPERIMENTAL RESULTS

The experimental results are presented from three aspects. First, the convergence results of DSPGD is shown in Figure 1 to verify its convergence rate. Second, the average communication cost per iteration of the two versions of DSPGD is provided in Figure 2 to show that CEM highly reduces the communication cost of DSPGD. Third, in Figure 3, FK $k$-means is compared with the cloud-based kernel $k$-means schemes in terms of clustering quality to show FK $k$-means can achieve the comparable clustering results as that of the cloud-based schemes; FK $k$-means is also compared with the existing distributed kernel $k$-means schemes under the federated settings to show the higher communication efficiency of FK $k$-means.

The convergence of DSPGD is validated over two datasets, Mushrooms and MNIST-small, whose low rank matrices $\hat{\mathbf{K}} = \sum_{i=1}^{s} \lambda_i \mathbf{u}_i \mathbf{u}_i^\top$ can be computed by performing SVD on their kernel matrices $\mathbf{K}$. A normalized recover error $\frac{\|\mathbf{K}_t - \hat{\mathbf{K}}\|_F^2}{N^2}$ (Zhang et al., 2016) is recorded for each iteration $t$ of DSPGD, where $\mathbf{K}_t = \sum_{i=1}^{s} (\tilde{\lambda}_{i,t} + (1 - \eta_t)\lambda)\tilde{\mathbf{u}}_{i,t}\tilde{\mathbf{u}}_{i,t}^\top$ is the estimation of $\hat{\mathbf{K}}$ at iteration $t$. In the left subfigure of Figure 1(a) and that of Figure 1(b), the convergence curves of DSPGD are lower than the curve of $0.4/t$ and the curve of $0.03/t$, respectively, which verifies DSPGD converges at an $O(1/t)$ rate. In Figure 1(a) and 1(b), the curves of two versions of DSPGD nearly overlap, indicating that CEM has little impact on the convergence of DSPGD.

The average communication cost per iteration of DSPGD with CEM and that of DSPGD without CEM are compared in Figure 2 to evaluate the effectiveness of CEM. The log-scale is used for the y-axis of each subfigure, and the unit of the y-axis is the number of the floating-point numbers. As shown in the four subfigures, CEM can reduce communication cost of DSPGD by more than 98%, which indicates that CEM is important for FK $k$-means to maintain communication efficiency.

In order to evaluate the clustering quality and the communication cost of FK $k$-means, curves of average normalized mutual information (NMI) (Strehl & Ghosh, 2002) versus average communi-

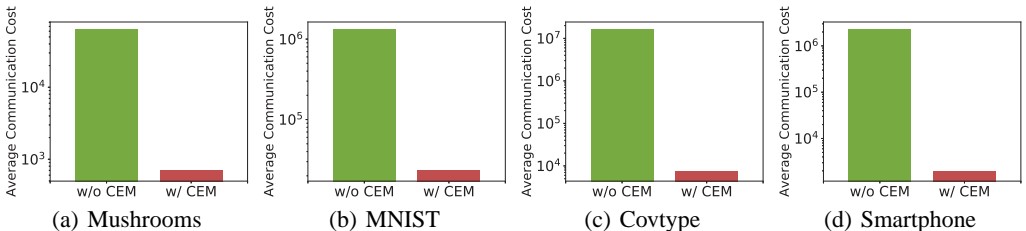

Figure 2: The average communication cost per iteration of the two versions of DSPGD.

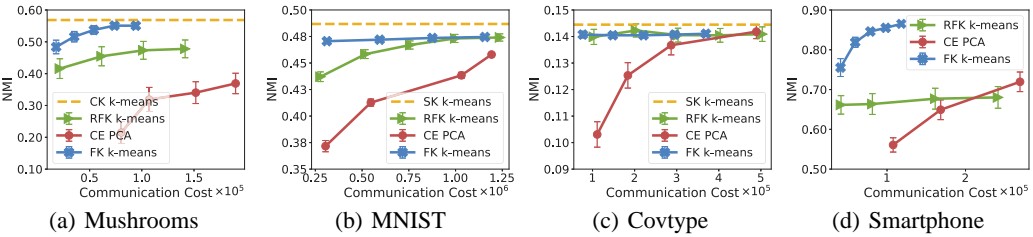

Figure 3: The NMI score versus the average communication cost of FK $k$-means and the existing methods on the four datasets.

cation cost are plotted in Figure 3 for FK $k$-means and the existing schemes. The error bars in Figure 3 are the $95\%$ confidence interval of the average NMI scores. For the three public datasets, FK $k$-means can achieve comparable average NMI scores to that of the cloud-based algorithms (CK $k$-means for Mushroom dataset and SK $k$-means for MNIST dataset and covtype dataset). For a cloud-based algorithm, its communication cost equals to the volume of a dataset, which is too large to be shown. Thus, in Figure 3(a), 3(b), and 3(c) the dash line only represents the average NMI scores rather than the relationship of average NMI scores versus the corresponding communication cost. In Figure 3(a) and 3(b), given a fixed communication cost, FK $k$-means can achieve the highest average NMI scores among the three client-based algorithms. Besides, in 3(b), FK $k$-means nearly achieves the upper bound of the average NMI score with a low communication cost. To reach such a NMI score, FK $k$-means reduces the communication cost by more than $60\%$ compared with RFK $k$-means. In Figure 3(c), FK $k$-means has a similar performance to that of RFK $k$-means. Compared with CE PCA, FK $k$-means reduces the communication cost by more than $60\%$ when the highest average NMI score is considered. For the Smartphone dataset, since it has no labels, the cluster quality of a given clustering algorithm is evaluated by measuring the similarity between the clustering results of the algorithm and that of SK $k$-means. To this end, the clustering results of SK $k$-means are used as the labels to compute NMI scores. It is shown in Figure 3(d) that FK $k$-means has a much higher upper bound for the average NMI score (close to $0.9$) than that of RFK $k$-means and CE PCA.

## 7 CONCLUSION

In this paper, FK $k$-means was developed. In the algorithm, a distributed stochastic proximal gradient descent approach was first designed to determine the eigenpairs of the kernel matrix in a distributed manner. A communication efficient mechanism was then designed to reduce the communication cost. In theoretical analysis, DSPGD with CEM was proved to converge at an $O(1/T)$ rate. The communication cost of DSPGD with CEM is unrelated to the number of data samples. The clustering loss of FK $k$-means can approach that of the centralized kernel $k$-means. It was also analyzed that FK $k$-means provided two levels of privacy preservation. The effectiveness of the FK $k$-means was validated by experiments on several real-world datasets. FK $k$-means can still be improved in terms of the asynchronous execution, the robustness to dropout users, and stronger privacy, which can be interesting topics in our future work.

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

---

**Algorithm 2** Lanczos Algorithm

---

1: **Input:** An symmetric matrix $\mathbf{R}$, an initial vector $\mathbf{c}_1$
2: **Output:** An approximation $\mathbf{P}_Q$ to the eigenvectors of $\mathbf{R}$, and an approximation $\boldsymbol{\sigma} = [\sigma_1, ..., \sigma_Q]$ to the eigenvalues of $\mathbf{R}$
3: Initialize $\beta_0 = 0$ and $\mathbf{c}_0 = \mathbf{0}$
4: **for** $q = 1, 2, \ldots, Q$ **do**
5:     $\mathbf{g} = \mathbf{R}\mathbf{c}_q$
6:     $\alpha_q = \mathbf{c}_q^\top \mathbf{g}$
7:     $\mathbf{g} = \mathbf{g} - \alpha_q \mathbf{c}_q - \beta_{q-1} \mathbf{c}_{q-1}$
8:     $\beta_q = ||\mathbf{g}||_2$
9:     **if** $\beta_q = 0$ **then**
10:         **break**
11:     **end if**
12:     $\mathbf{c}_{q+1} = \mathbf{g}/\beta_q$
13:     Construct a symmetric tridiagonal matrix $\mathbf{T}_Q$
14:     Perform EVD on $\mathbf{T}_Q$ to obtain its eigenvectors $\mathbf{P}_Q$ and its eigenvalues $\boldsymbol{\sigma} = [\sigma_1, ..., \sigma_Q]$
15:     Compute $\mathbf{C}_Q \mathbf{P}_Q$
16: **end for**

---

## A  DETAILS OF RANDOM FEATURE METHOD

For a kernel matrix $\mathbf{K}$, a random feature method (Rahimi & Recht, 2008; Kar & Karnick, 2012) can generate an unbiased estimate of $\mathbf{K}$, denoted as $\boldsymbol{\xi}$, with the following expression:

$$\boldsymbol{\xi} = \frac{1}{D}\mathbf{A}\mathbf{A}^\top,$$

where the $i$-th row of $\mathbf{A}$ is the random feature vectors $\mathbf{a}(\mathbf{x}_i)$ for the data sample $\mathbf{x}_i$. The matrix $\boldsymbol{\xi}$ satisfies $\mathbb{E}[\boldsymbol{\xi}] = \mathbf{K}$.

We then use the example of shift-invariant kernels to show how a random feature vector is constructed. For popular shift-invariant kernels $\kappa(\mathbf{x}_i, \mathbf{x}_j)$ with Fourier representation

$$\kappa(\mathbf{x}_i, \mathbf{x}_j) = \int p(\mathbf{w})\exp(j\mathbf{w}^\top(\mathbf{x}_i - \mathbf{x}_j))d\mathbf{w}$$

where $p(\mathbf{w})$ is a probability density function, they can be estimated by the random Fourier features (Rahimi & Recht, 2008) as follows. By randomly drawing $D$ independent samples $\{\mathbf{w}_1, \ldots, \mathbf{w}_D\}$ from $p(\mathbf{w})$, a random feature vector $\mathbf{a}(\mathbf{x}_i)$ for a data sample $\mathbf{x}_i$ can be written as $\mathbf{a}(\mathbf{x}_i) = [\sqrt{2}\cos(\mathbf{w}_1^\top \mathbf{x}_i + b_1), \ldots, \sqrt{2}\cos(\mathbf{w}_D^\top \mathbf{x}_i + b_D)]^\top$ where $\{b_1, \ldots, b_D\}$ are independent random variables drawn from $[0, 2\pi)$ uniformly. As a result, an unbiased estimation of $\mathbf{K}$ can be written as $\boldsymbol{\xi} = \frac{1}{D}\mathbf{A}\mathbf{A}^\top$ where $\mathbf{A} = [\mathbf{a}(\mathbf{x}_1), \cdots, \mathbf{a}(\mathbf{x}_n)]^\top$.

## B  DETAILS ABOUT DISTRIBUTED LANCZOS ALGORITHM

To find the eigenpairs of a symmetric matrix $\mathbf{R}$, the Lanczos algorithm (LA) (Lanczos, 1950) first build a Krylov subspace $\mathcal{K}_q(\mathbf{R}, \mathbf{c}_1) = \mathrm{span}[\mathbf{c}_1, \mathbf{R}\mathbf{c}_1, ..., \mathbf{R}^{q-1}\mathbf{c}_1]$ where $\mathbf{c}_1$ is an initial vector, and then it employs the Rayliegh-Ritz procedure to construct the best approximate eigenpairs for $\mathbf{R}$ in the Krylov subspace. In the first step, LA constructs an orthogonal basis of the Krylov subspace following the procedure of line 5 to line 12 in Algorithm 2. Meanwhile, a symmetric tridiagonal matrix $\mathbf{T}_Q = \mathbf{C}_Q^\top \mathbf{R}\mathbf{C}_Q$ can be explicitly constructed with $\boldsymbol{\alpha}_Q$ and $\boldsymbol{\beta}_Q$ via

$$\mathbf{T}_Q = \begin{bmatrix} \alpha_1 & \beta_1 & & \\ \beta_1 & \ddots & \ddots & \\ & \ddots & \ddots & \beta_{Q-1} \\ & & \beta_{Q-1} & \alpha_Q \end{bmatrix}.$$

Based on $\mathbf{T}_Q$, the Rayliegh-Ritz procedure can be utilized to approximate the eigenpairs of $\mathbf{R}$. Let $\mathbf{T}_Q = \mathbf{P}_Q \boldsymbol{\Sigma}_Q \mathbf{P}_Q^\top$ be the eigendecomposition of $\mathbf{T}_Q$. It has been proved that the columns of $\mathbf{C}_Q \mathbf{P}_Q$

and the diagonal entries of $\mathbf{\Sigma}_Q$ are the optimal approximation to the eigenvectors and eigenvalues of $\mathbf{R}$, respectively (Demmel, 1997). Thus, in the Rayliegh-Ritz procedure, $\mathbf{P}_Q$ and $\mathbf{\Sigma}_Q$ are determined by performing EVD on $\mathbf{T}_Q$, and then $\mathbf{C}_Q\mathbf{P}_Q$ are computed as the approximation to the eigenvectors of $\mathbf{R}$. As the number of iteration $Q$ increases, the columns of $\mathbf{C}_Q\mathbf{P}_Q$ and the diagonal entries of $\mathbf{\Sigma}_Q$ can converge to the eigenvectors and eigenvalues of $\mathbf{R}$, respectively (Demmel, 1997).

As for the distributed Lanczos algorithm (DLA), only the step of line 5 in Algorithm 2 is conducted in a distributed manner, and other steps are conducted at the cloud server. In our problem, if

$$\mathbf{R} = (1 - \eta_t)\mathbf{Z}_t + \eta_t\boldsymbol{\xi}_t = (1 - \eta_t)\mathbf{Z}_t + \frac{\eta_t}{D}\mathbf{A}_t\mathbf{A}_t^\top,$$

where $\mathbf{Z}_t$ and $\mathbf{c}_q$ are known at the cloud server, and $\mathbf{A}_t = [\mathbf{A}_t[1]^\top, \dots, \mathbf{A}_t[M]^\top]^\top$ are distributed over $M$ users' devices, then $(1 - \eta_t)\mathbf{Z}_t\mathbf{c}_q$ is computed at the cloud server and $\eta_t\mathbf{A}_t\mathbf{A}_t^\top\mathbf{c}_q/D$ is computed in a distributed manner as follows. The vector $\mathbf{c}_q = [\mathbf{c}_q[1]^\top, \dots, \mathbf{c}_q[M]^\top]^\top$ is first partitioned into $M$ parts at the cloud server, and the $m$-th part $\mathbf{c}_q[m]$ is sent to the $m$-th user's device. A local vector $\mathbf{A}_t[m]^\top\mathbf{c}_q[m]$ is then computed at the $m$-th user's device. These local vectors from $M$ users' devices are summed up at the cloud to obtain a vector $\mathbf{A}_t^\top\mathbf{c}_q$. $\mathbf{A}_t^\top\mathbf{c}_q$ is then broadcast to $M$ users' devices, and a vector $\mathbf{A}_t[m]\mathbf{A}_t^\top\mathbf{c}_q$ is computed at the $m$-th user's device. These $M$ vectors are sent back to the cloud server where they are concatenated to form $\mathbf{A}_t\mathbf{A}_t^\top\mathbf{c}_q$. If

$$\mathbf{R} = \mathbf{W}_t^\top\mathbf{W}_t = \sum_{m=1}^{M} \mathbf{W}_t[m]^\top\mathbf{W}_t[m],$$

where $\mathbf{W}_t[m]^\top\mathbf{W}_t[m]$ can be computed at the $m$-th user's device, then each user' devices first determines $\mathbf{W}_t[m]^\top\mathbf{W}_t[m]\mathbf{c}_q$ locally, and these $M$ vectors are then uploaded to the cloud server where they are summed up to form $\mathbf{W}_t^\top\mathbf{W}_t\mathbf{c}_q$.

In the $t$-th iteration of DSPGD, DLA is used to compute the eigenvalues larger than $\eta_t\lambda$ of the $\mathbf{R}_t$. Thus, in practice, the convergence criterion of DLA is that all the approximated eigenvalues larger than $\eta_t\lambda$ converge, rather than that the number of iteration reaches its maximal value $Q$.

One issue of LA and DLA in practice is that they can only be conducted in floating point arithmetic, which can destroy the orthogonality of the columns in $\mathbf{C}_q$, and further affect the convergence of DSPGD. To this end, a full reorthogonalization method (Demmel, 1997) is utilized to guarantee that $\mathbf{C}_q$ is an orthogonal matrix with a high probability. The key idea of this method to generate a new vector $\mathbf{c}_q$ from a subspace that is orthogonal to all the previous vectors $\{\mathbf{c}_1, \dots, \mathbf{c}_{q-1}\}$, which can be accomplished by replacing line 7 in Algorithm 2 with

$$\mathbf{g} = \mathbf{g} - \sum_{i=1}^{q-1} \mathbf{g}^\top\mathbf{c}_i\mathbf{c}_i. \tag{3}$$

The operation in (3) can be called multiple times in one iteration of LA to increase the probability that $\mathbf{C}_q$ is an orthogonal matrix. In the implementation of federated kernel $k$-means, such operation is called twice in each iteration of DLA. Note that the full reorthogonalization only requires more flops at the cloud server, which does not affect the algorithm complexity at users' devices.

## C  PROOF OF THEOREM 1

This proof partially follows the proof of Theorem 1 in Zhang et al. (2016). The difference is that the $t$-th iteration of $\mathbf{Z}^*$ obtained by DSPGD, i.e., $\mathbf{Z}_t$, may not equal $\mathbf{Z}_t^* = \mathcal{D}_{\eta_t\lambda}[(1 - \eta_t)\mathbf{Z}_t + \eta_t\boldsymbol{\xi}_t]$. The gap between $\mathbf{Z}_t$ and $\mathbf{Z}_t^*$ is caused by that the distributed Lanczos algorithm (DLA) only approximates the eigenpairs of a target matrix. Thus, in this proof, it is assumed that

$$||\mathbf{Z}_t - \mathbf{Z}_t^*||_F^2 \leq \epsilon, \quad \forall t, \tag{4}$$

when DLA reaches its convergence criterion, where $\epsilon \ll 1$.

Before the proof, we first define

$$F(\mathbf{Z}) = \frac{1}{2}\mathbb{E}[||\mathbf{Z} - \boldsymbol{\xi}||_F^2],$$

$$f_t(\mathbf{Z}) = \frac{1}{2}||\mathbf{Z} - \boldsymbol{\xi}_t||_F^2.$$

For a $\mu$-strongly convex function $l(\mathbf{Z})$, if $l(\mathbf{Z}_1) \geq l(\mathbf{Z}_2)$, then

$$l(\mathbf{Z}_1) - l(\mathbf{Z}_2) \geq \frac{\mu}{2}||\mathbf{Z}_1 - \mathbf{Z}_2||_F^2. \tag{5}$$

In the $t+1$-th iteration of DSPGD, the goal is to determine the optimal solution $\mathbf{Z}_{t+1}^*$ to the following optimization problem

$$\min_{\mathbf{Z} \in \mathbb{R}^{n \times n}} \frac{1}{2}||\mathbf{Z} - \mathbf{Z}_t||_F^2 + \eta_t \langle \mathbf{Z} - \mathbf{Z}_t, \nabla f_t(\mathbf{Z}_t) \rangle + \eta \lambda ||\mathbf{Z}||_*. \tag{6}$$

By DLA, an approximate solution $\mathbf{Z}_{t+1}$ that satisfies (4) can be obtained. The following lemma is a key step in this proof.

**Lemma 1.** *Before the convergence of DSPGD, the following inequality holds, i.e.,*

$$\frac{1}{2}||\mathbf{Z}_{t+1} - \mathbf{Z}_t||_F^2 + \eta_t \langle \mathbf{Z}_{t+1} - \mathbf{Z}_t, \nabla f_t(\mathbf{Z}_t) \rangle + \eta \lambda ||\mathbf{Z}_{t+1}||_*$$
$$\leq \frac{1}{2}||\mathbf{Z}^* - \mathbf{Z}_t||_F^2 + \eta_t \langle \mathbf{Z}^* - \mathbf{Z}_t, \nabla f_t(\mathbf{Z}_t) \rangle + \eta \lambda ||\mathbf{Z}^*||_*. \tag{7}$$

*Proof.* The objective function in (6) can be rewritten as

$$\frac{1}{2}||\mathbf{Z} - \mathbf{Z}_t||_F^2 + \eta_t \langle \mathbf{Z} - \mathbf{Z}_t, \nabla f_t(\mathbf{Z}_t) \rangle + \eta \lambda ||\mathbf{Z}||_*$$
$$= \frac{1}{2}||\mathbf{Z} - \mathbf{Z}_t||_F^2 + \eta_t \langle \mathbf{Z} - \mathbf{Z}_t, \nabla f_t(\mathbf{Z}_t) \rangle + \frac{\eta_t^2}{2}||\nabla f_t(\mathbf{Z}_t)||_F^2 - \frac{\eta_t^2}{2}||\nabla f_t(\mathbf{Z}_t)||_F^2 + \eta \lambda ||\mathbf{Z}||_*$$
$$= \frac{1}{2}||\mathbf{Z} - [(1 - \eta_t)\mathbf{Z}_t + \eta_t \boldsymbol{\xi}]||_F^2 + \eta \lambda ||\mathbf{Z}||_* - \frac{\eta_t^2}{2}||\nabla f_t(\mathbf{Z}_t)||_F^2.$$

Since $\frac{\eta_t^2}{2}||\nabla f_t(\mathbf{Z}_t)||_F^2$ is a constant, we can only consider

$$l(\mathbf{Z}) = \frac{1}{2}||\mathbf{Z} - [(1 - \eta_t)\mathbf{Z}_t + \eta_t \boldsymbol{\xi}]||_F^2 + \eta \lambda ||\mathbf{Z}||_*$$

in the following part of the proof. Now we first assume that $l(\mathbf{Z}^*) \leq l(\mathbf{Z}_{t+1})$, then we have

$$l(\mathbf{Z}_{t+1}) - l(\mathbf{Z}_{t+1}^*) \geq l(\mathbf{Z}^*) - l(\mathbf{Z}_{t+1}^*) \geq \frac{\mu}{2}||\mathbf{Z}_{t+1}^* - \mathbf{Z}^*||_F^2. \tag{8}$$

Moreover, $l(\mathbf{Z}_{t+1}) - l(\mathbf{Z}_{t+1}^*)$ can be expanded as

$$l(\mathbf{Z}_{t+1}) - l(\mathbf{Z}_{t+1}^*)$$
$$= \frac{1}{2}||\mathbf{Z}_{t+1} - \mathbf{R}_t||_F^2 + \eta \lambda ||\mathbf{Z}_{t+1}||_* - \frac{1}{2}||\mathbf{Z}_{t+1}^* - \mathbf{R}_t||_F^2 + \eta \lambda ||\mathbf{Z}_{t+1}^*||_*$$
$$= \frac{1}{2}(||\mathbf{Z}_{t+1} - \mathbf{R}_t||_F - ||\mathbf{Z}_{t+1}^* - \mathbf{R}_t||_F)(||\mathbf{Z}_{t+1} - \mathbf{R}_t||_F + ||\mathbf{Z}_{t+1}^* - \mathbf{R}||_F) \tag{9}$$
$$\quad + \eta_t \lambda (||\mathbf{Z}_{t+1}||_* - ||\mathbf{Z}_{t+1}^*||_*)$$
$$\leq \frac{1}{2}||\mathbf{Z}_{t+1} - \mathbf{Z}_{t+1}^*||_F (||\mathbf{Z}_{t+1} - \mathbf{Z}_{t+1}^*||_F + 2||\mathbf{Z}_{t+1}^* - \mathbf{R}||_F) + \eta_t \lambda ||\mathbf{Z}_{t+1} - \mathbf{Z}_{t+1}^*||_*$$

It is well known that given a matrix $\mathbf{M}$ the following inequality holds for its nuclear norm and its Frobenius norm, i.e., $||\mathbf{M}||_*^2 \leq \text{rank}(\mathbf{M})||\mathbf{M}||_F^2$. By this inequality, we have

$$||\mathbf{Z}_{t+1} - \mathbf{Z}_{t+1}^*||_* \leq \sqrt{r}||\mathbf{Z}_{t+1} - \mathbf{Z}_{t+1}^*||_F \leq \sqrt{r}\epsilon, \tag{10}$$

where $r$ is the rank of $(\mathbf{Z}_{t+1} - \mathbf{Z}_{t+1}^*)$. Substitute (4) and (10) into (9), we have

$$l(\mathbf{Z}_{t+1}) - l(\mathbf{Z}_{t+1}^*) \leq \frac{1}{2}\epsilon^2 + \epsilon||\mathbf{Z}_{t+1}^* - \mathbf{R}_t||_F + \eta_t \lambda \sqrt{r}\epsilon.$$

Since $||\mathbf{Z}_{t+1}^* - \mathbf{R}_t||_F$ is a constant, this upper bound of $l(\mathbf{Z}_{t+1}) - l(\mathbf{Z}_{t+1}^*)$ can become arbitrarily small if $\epsilon$ is arbitrarily small. Hence, according to (8), $||\mathbf{Z}_{t+1}^* - \mathbf{Z}^*||_F^2$ can also be arbitrarily small. However, this contradicts that $||\mathbf{Z}_{t+1}^* - \mathbf{Z}^*||_F^2$ cannot become arbitrarily small before the convergence of DSPGD. Therefore, before the convergence of DSPGD, $l(\mathbf{Z}^*) \geq l(\mathbf{Z}_{t+1})$ is satisfied. $\qquad\square$

The rest part then follows the proof of Theorem 1 in Zhang et al. (2016). Based on Lemma 1 and the property of strongly convex function in (5), the update rule of SPDG implies

$$
\begin{aligned}
&\frac{1}{2}||\mathbf{Z}_{t+1} - \mathbf{Z}_t||_F^2 + \eta_t\langle\mathbf{Z}_{t+1} - \mathbf{Z}_t, \nabla f_t(\mathbf{Z}_t)\rangle + \eta\lambda||\mathbf{Z}_{t+1}||_* \\
&\leq\frac{1}{2}||\mathbf{Z}^* - \mathbf{Z}_t||_F^2 + \eta_t\langle\mathbf{Z}^* - \mathbf{Z}_t, \nabla f_t(\mathbf{Z}_t)\rangle + \eta_t\lambda||\mathbf{Z}^*||_* - \frac{1}{2}||\mathbf{Z}^* - \mathbf{Z}_{t+1}||_F^2.
\end{aligned}
\tag{11}
$$

Since $F(\mathbf{Z})$ is 1-strongly convex, it can be shown that

$$
\begin{aligned}
&\frac{1}{2}||\mathbf{Z}_t - \mathbf{Z}^*||_F^2 \\
&\leq F(\mathbf{Z}_t) + \lambda||\mathbf{Z}_t||_* - F(\mathbf{Z}^*) - \lambda||\mathbf{Z}^*||_* \\
&\leq\langle\mathbf{Z}_t - \mathbf{Z}^*, \nabla F(\mathbf{Z}_t)\rangle - \frac{1}{2}||\mathbf{Z}_t - \mathbf{Z}^*||_F^2 + \lambda||\mathbf{Z}_t||_* - \lambda||\mathbf{Z}^*||_* \\
&=\langle\mathbf{Z}_t - \mathbf{Z}^*, \nabla f_t(\mathbf{Z}_t)\rangle - \lambda||\mathbf{Z}^*||_* - \frac{1}{2\eta_t}||\mathbf{Z}_t - \mathbf{Z}^*||_F^2 \\
&\quad + \lambda||\mathbf{Z}_t||_* - \frac{1}{2}||\mathbf{Z}_t - \mathbf{Z}^*||_F^2 + \frac{1}{2\eta_t}||\mathbf{Z}_t - \mathbf{Z}^*||_F^2 + \langle\nabla F(\mathbf{Z}_t) - \nabla f_t(\mathbf{Z}_t), \mathbf{Z}_t - \mathbf{Z}^*\rangle \\
&\overset{(11)}{\leq}\langle\mathbf{Z}_t - \mathbf{Z}_{t+1}, \nabla f_t(\mathbf{Z}_t)\rangle - \lambda||\mathbf{Z}_{t+1}||_* - \frac{1}{2\eta_t}||\mathbf{Z}_{t+1} - \mathbf{Z}_t||_F^2 - \frac{1}{2\eta_t}||\mathbf{Z}^* - \mathbf{Z}_{t+1}||_F^2 \\
&\quad + \lambda||\mathbf{Z}_t||_* + \frac{1}{2}\left(\frac{1}{\eta_t} - 1\right)||\mathbf{Z}_t - \mathbf{Z}^*||_F^2 + \langle\nabla F(\mathbf{Z}_t) - \nabla f_t(\mathbf{Z}_t), \mathbf{Z}_t - \mathbf{Z}^*\rangle \\
&\leq\max_{\mathbf{W}}\left(\langle\mathbf{W}, \nabla f_t(\mathbf{Z}_t)\rangle - \frac{1}{2\eta_t}||\mathbf{W}||_F^2\right) - \frac{1}{2\eta_t}||\mathbf{Z}_{t+1} - \mathbf{Z}^*||_F^2 \\
&\quad + \lambda||\mathbf{Z}_t||_* - \lambda||\mathbf{Z}_{t+1}||_* + \frac{1}{2}\left(\frac{1}{\eta_t} - 1\right)||\mathbf{Z}_t - \mathbf{Z}^*||_F^2 + \langle\nabla F(\mathbf{Z}_t) - \nabla f_t(\mathbf{Z}_t), \mathbf{Z}_t - \mathbf{Z}^*\rangle \\
&=\frac{\eta_t}{2}||\nabla f_t(\mathbf{Z}_t)||_F^2 - \frac{1}{2\eta_t}||\mathbf{Z}_{t+1} - \mathbf{Z}^*||_F^2 \\
&\quad + \lambda||\mathbf{Z}_t||_* - \lambda||\mathbf{Z}_{t+1}||_* + \frac{1}{2}\left(\frac{1}{\eta_t} - 1\right)||\mathbf{Z}_t - \mathbf{Z}^*||_F^2 + \langle\nabla F(\mathbf{Z}_t) - \nabla f_t(\mathbf{Z}_t), \mathbf{Z}_t - \mathbf{Z}^*\rangle,
\end{aligned}
\tag{12}
$$

where the third inquality holds based on the inequality in (11).

By substituting $\delta_t = \langle\boldsymbol{\xi}_t - \mathbf{K}, \mathbf{Z}_t - \mathbf{Z}^*\rangle$ and $C^2 = \max_{t\in[T]}||\mathbf{Z}_t - \boldsymbol{\xi}_t||_F^2$ into (12),

$$
||\mathbf{Z}_{t+1} - \mathbf{Z}^*||_F^2 \leq \eta_t^2 C^2 + 2\eta_t\delta_t + 2\lambda\eta_t\left(||\mathbf{Z}_t||_* - ||\mathbf{Z}_{t+1}||_*\right) + (1 - 2\eta_t)||\mathbf{Z}_t - \mathbf{Z}^*||_F^2. \tag{13}
$$

The inequality in (13) is the same as the result of Lemma 1 in Zhang et al. (2016). Thus, the following lemmas [2] from Zhang et al. (2016) can be directly utilized to derive a probability bound for $||\mathbf{Z}_{t+1} - \mathbf{Z}^*||_F^2$.

**Lemma 2** (Lemma 2 in Zhang et al. (2016)). *Define $\gamma = \max_{t\in[T]}||\mathbf{Z}_t||_*$. By setting $\eta_t = \frac{2}{t}$, an upper bound of $||\mathbf{Z}_{t+1} - \mathbf{Z}^*||_F^2$ can be written as*

$$
||\mathbf{Z}_{T+1} - \mathbf{Z}^*||_F^2 \leq \frac{4(C^2 + \lambda\gamma)}{T} + \frac{2}{T(T-1)}\left[2\sum_{t=2}^T(t-1)\delta_t - \sum_{t=2}^T(t-1)||\mathbf{Z}_t - \mathbf{Z}^*||_F^2\right].
$$

The upper bound of $\sum_{t=2}^T(t-1)\delta_t$ in Lemma 2 is then provided in Lemma 3.

**Lemma 3** (Lemma 3 in Zhang et al. (2016)). *Assume $||\boldsymbol{\xi}_t - \mathbf{K}||_F \leq G$, and $||\mathbf{Z}_t - \mathbf{Z}^*||_F \leq H$, $\forall t > 2$. With a probability at least $1 - \delta$, $\sum_{t=2}^T(t-1)\delta_t$ is upper bounded by*

$$
\sum_{t=2}^T(t-1)\delta_t \leq \frac{1}{2}\sum_{t=2}^T(t-1)||\mathbf{Z}_t - \mathbf{Z}^*||_F^2 + 2G^2\tau(T-1) + \frac{2}{3}GH(T-1)\tau + GH(T-1),
$$

---

[2]These lemmas can be found in the supplementary material of Zhang et al. (2016) that can be downloaded from `https://cs.nju.edu.cn/zlj/pdf/AAAI-2016-Zhang-S.pdf`

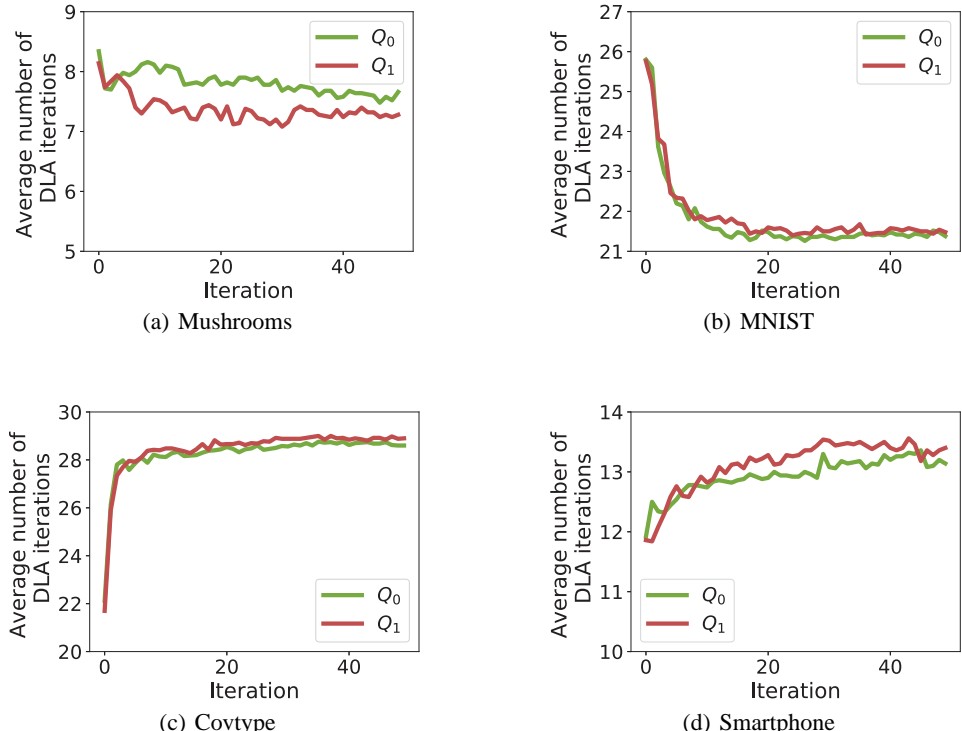

Figure 4: The values of $Q_0$ and $Q_1$ versus the number of iteration $t$ for the four real-world datasets

*where* $\tau = \log \frac{\lceil 2\log_2 T \rceil}{\delta}$.

Based on Lemma 2 and Lemma 3, the following upper bound of $||\mathbf{Z}_{T+1} - \mathbf{Z}^*||_F^2$ holds with a probability at least $1 - \delta$

$$||\mathbf{Z}_{T+1} - \mathbf{Z}^*||_F^2 \leq \frac{4}{T}\left(C^2 + \lambda\gamma + 2G^2\tau + \frac{2}{3}GH\tau + GH\right) = O(1/T). \tag{14}$$

## D    PROOF OF THEOREM 2 AND EMPIRICAL RESULTS

For DSPGD with CEM, in the $t$-the iteration, the $m$-th user's device only needs to upload one vector in each iteration of DLA, i.e., $\mathbf{W}_t[m]^\top\mathbf{W}_t[m]\mathbf{c}_q$. Since $\mathbf{W}_t[m] = [\sqrt{\frac{\eta_t}{D}}\mathbf{A}_t[m], \sqrt{1-\eta_t}\mathbf{B}_t[m]]$, the dimension of $\mathbf{W}_t[m]^\top\mathbf{W}_t[m]\mathbf{c}_q$ equals $D + r_t$, where $r_t$ is the rank of $\mathbf{Z}_t$ and $D$ is the number of random feature. Moreover, DLA requires several Lanczos iterations to approach the eigenpairs of $\mathbf{W}_t^\top\mathbf{W}_t$. Thus, the communication cost of DSPGD with CEM is linear to $D + r_t$.

To compute the ratio, we first derive the communication cost for DSPGD without CEM. in the $t$-the iteration, the $m$-th user's device needs to upload two vectors in each iteration of DLA: $\mathbf{A}_t[m]^\top\mathbf{c}_q[m]$ and $\mathbf{A}_t[m]\mathbf{A}_t^\top\mathbf{c}_q$, where the dimension of $\mathbf{A}_t^\top\mathbf{c}_q$ equals $D$. For the concatenation of all $M$ vectors $\{\mathbf{A}_t[m]\mathbf{A}_t^\top\mathbf{c}_q, m = 1,...,M\}$, its dimension equals the number of data samples $N$. Thus, the communication cost of DSPGD without CEM is linear to $N + MD$.

Given the number of Lanczos iterations $Q_0$ for DSPGD without CEM and the number of Lanczos iterations $Q_1$ for DSPGD with CEM, the ratio can be determined by $\frac{(N+MD)Q_0}{M(r_t+D)Q_1}$.

According to Figure 4, it can be seen that the average value of $Q_0$ is close to that of $Q_1$ in each iteration $t$, which indicates $\frac{Q_0}{Q_1} \approx 1$. As a result, the dominant factor of the ratio is still $\frac{N+MD}{M(r_t+D)}$

Empirically, the figures of the average value of $r_t$ versus the number of iterations $t$ for the four real-world datasets are shown in Figure 5. The results show that the rank $r_t$ tends to converge as

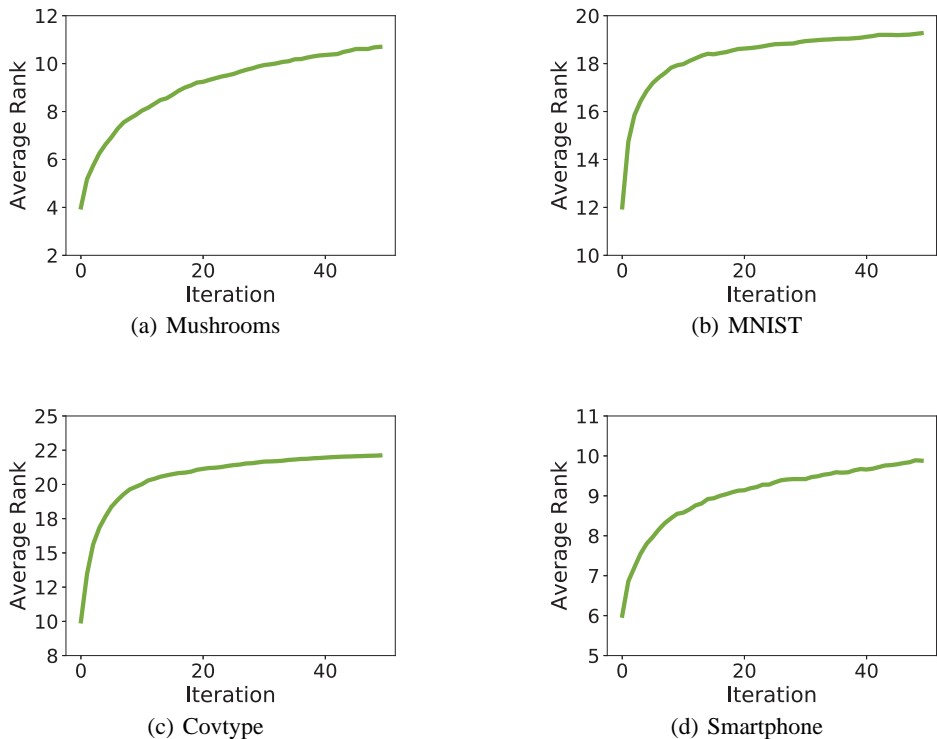

Figure 5: The rank of $\mathbf{Z}_t$ versus the number of iteration $t$ for the four real-world datasets

the value of $t$ increases. Besides, the upper bound of $r_t$ is a constant factor larger than the number of eigenvectors $s$ in Table 1, and such upper bound is much smaller than the number of users' local data samples, which can explain the dramatical reduction on the communication cost in Figure 2.

## E   PROOF OF THEOREM 3

Define $\mathbf{K} = \mathbf{U}\mathbf{\Lambda}\mathbf{U}^\top$, and $\mathbf{P} = \mathbf{U}\mathbf{\Lambda}^{\frac{1}{2}}$. The low-rank approximation of $\mathbf{K}$ with rank $s$ is denoted as $\mathbf{K}_s = \mathbf{U}\mathbf{\Lambda}_s\mathbf{U}^\top$, and $\mathbf{P}_s = \mathbf{U}\mathbf{\Lambda}_s^{\frac{1}{2}}$ where the diagonal of $\mathbf{\Lambda}_s$ contains the $s$ largest eigenvalues of $\mathbf{K}$ while its rest diagonal entries are all zero. The output of DSPGD at iteration $t$ is an estimation of $\mathbf{K}_s$, denoted as $\widetilde{\mathbf{K}}_t$, and $\widetilde{\mathbf{K}}_t = \widetilde{\mathbf{P}}_t\widetilde{\mathbf{P}}_t^\top$.

The following two lemmas will be used in the proof of Theorem 3.

**Lemma 4.** *Given* $\widetilde{\mathbf{K}}_t$*, the following inequality holds with a probability at least* $1 - \delta$ *for any rank* $k$ *projection matrix* $\mathbf{\Pi} \in \mathbb{R}^{n \times n}$*,*

$$\mathrm{Tr}(\mathbf{\Pi}(\mathbf{K}_s - \widetilde{\mathbf{K}}_t)) \leq O(\sqrt{\frac{s}{t}})$$

*Proof.* Since $\mathbf{\Pi}$ is a rank-$k$ projection matrix, it is obvious that

$$\mathrm{Tr}(\mathbf{\Pi}(\mathbf{K}_s - \widetilde{\mathbf{K}}_t)) \leq ||\mathbf{K}_s - \widetilde{\mathbf{K}}_t||_*.$$

For a rank-$s$ matrix $\mathbf{A}$, the following inequality holds for its Nuclear norm and its Frobenius norm

$$||\mathbf{A}||_*^2 \leq s||\mathbf{A}||_F^2.$$

Hence,

$$||\mathbf{K}_s - \widetilde{\mathbf{K}}_t||_* \leq \sqrt{s}||\mathbf{K}_s - \widetilde{\mathbf{K}}_t||_F.$$

By Theorem 1, $\mathbf{Z}_t$ converges to $\mathbf{Z}^*$ at an $O(1/t)$ rate. Note $\mathbf{Z}^*$ has the same eigenvectors as $\mathbf{K}_s$. Thus, $\widetilde{\mathbf{K}}_t$ constructed based on $\mathbf{Z}_t$ also converges to $\mathbf{K}_s$ at an $O(1/t)$ rate with a probability at least $1 - \delta$, i.e., $||\mathbf{K}_s - \widetilde{\mathbf{K}}_t||_F^2$ has an upper bound as

$$||\mathbf{K}_s - \widetilde{\mathbf{K}}_t||_F^2 \leq O(1/t).$$

Hence, the following inequality holds with a probability at least $1 - \delta$

$$\mathrm{Tr}(\mathbf{\Pi}(\mathbf{K}_s - \widetilde{\mathbf{K}}_t)) \leq \sqrt{s}||\mathbf{K}_s - \widetilde{\mathbf{K}}_t||_F$$
$$\leq O(\sqrt{\frac{s}{t}}).$$

$\square$

**Lemma 5.** *Fix an error parameter $\varepsilon \in (0, 1)$. For any rank $k$ projection matrix $\mathbf{\Pi} \in \mathbb{R}^{n \times n}$,*

$$\mathrm{Tr}\left(\mathbf{\Pi}(\mathbf{K} - \widetilde{\mathbf{K}}_t)\mathbf{\Pi}\right) \leq (\varepsilon + \frac{k}{s})||\mathbf{P} - \mathbf{\Pi}\mathbf{P}||_F^2.$$

*Proof.* It holds that

$$\mathrm{Tr}\left((\mathbf{I}_n - \mathbf{\Pi})(\mathbf{K} - \widetilde{\mathbf{K}}_t)\right) = \mathrm{Tr}(\mathbf{K} - \widetilde{\mathbf{K}}_t) - \mathrm{Tr}\left(\mathbf{\Pi}(\mathbf{K} - \widetilde{\mathbf{K}}_t)\mathbf{\Pi}\right)$$
$$= ||\mathbf{K} - \mathbf{K}_s||_* + \mathrm{Tr}(\mathbf{K}_s - \widetilde{\mathbf{K}}_t) - \mathrm{Tr}\left(\mathbf{\Pi}(\mathbf{K} - \widetilde{\mathbf{K}}_t)\mathbf{\Pi}\right).$$

Thus, $\mathrm{Tr}\left(\mathbf{\Pi}(\mathbf{K} - \widetilde{\mathbf{K}}_t)\mathbf{\Pi}\right)$ can be rewritten as

$$\mathrm{Tr}\left(\mathbf{\Pi}(\mathbf{K} - \widetilde{\mathbf{K}}_t)\mathbf{\Pi}\right) = ||\mathbf{K} - \mathbf{K}_s||_* + \mathrm{Tr}(\mathbf{K}_s - \widetilde{\mathbf{K}}_t) - \mathrm{Tr}\left((\mathbf{I}_n - \mathbf{\Pi})(\mathbf{K} - \widetilde{\mathbf{K}}_t)(\mathbf{I}_n - \mathbf{\Pi})\right).$$

It follows that

$$\mathrm{Tr}\left((\mathbf{I}_n - \mathbf{\Pi})(\mathbf{K} - \widetilde{\mathbf{K}}_t)(\mathbf{I}_n - \mathbf{\Pi})\right)$$
$$= \mathrm{Tr}\left((\mathbf{I}_n - \mathbf{\Pi})(\mathbf{K} - \mathbf{K}_s)(\mathbf{I}_n - \mathbf{\Pi})\right) + \mathrm{Tr}\left((\mathbf{I}_n - \mathbf{\Pi})(\mathbf{K}_s - \widetilde{\mathbf{K}}_t)(\mathbf{I}_n - \mathbf{\Pi})\right)$$
$$= \mathrm{Tr}\left((\mathbf{I}_n - \mathbf{\Pi})(\mathbf{K} - \mathbf{K}_s)(\mathbf{I}_n - \mathbf{\Pi})\right) + \mathrm{Tr}(\mathbf{K}_s - \widetilde{\mathbf{K}}_t) - \mathrm{Tr}(\mathbf{\Pi}(\mathbf{K}_s - \widetilde{\mathbf{K}}_t))$$
$$\geq ||\mathbf{P} - \mathbf{P}_{s+k}||_F^2 + \mathrm{Tr}(\mathbf{K}_s - \widetilde{\mathbf{K}}_t) - O(\sqrt{\frac{s}{t}}),$$

where the last inequality comes from Lemma 4. Thus,

$$\mathrm{Tr}\left(\mathbf{\Pi}(\mathbf{K} - \widetilde{\mathbf{K}}_t)\mathbf{\Pi}\right) \leq ||\mathbf{K} - \mathbf{K}_s||_* - ||\mathbf{P} - \mathbf{P}_{s+k}||_F^2 + O(\sqrt{\frac{s}{t}})$$
$$= ||\mathbf{P} - \mathbf{P}_s||_F^2 - ||\mathbf{P} - \mathbf{P}_{s+k}||_F^2 + O(\sqrt{\frac{s}{t}})$$
$$= \sum_{i=s+1}^{n} \sigma_i^2(\mathbf{P}) - \sum_{i=s+k+1}^{n} \sigma_i^2(\mathbf{P}) + O(\sqrt{\frac{s}{t}})$$
$$= \sum_{i=s+1}^{s+k} \sigma_i^2(\mathbf{P}) + O(\sqrt{\frac{s}{t}})$$
$$\leq \frac{k}{s} \sum_{i=k+1}^{s+k} \sigma_i^2(\mathbf{P}) + O(\sqrt{\frac{s}{t}}).$$

Since $O(\sqrt{\frac{s}{t}})$ can be arbitrarily small, it can be rewritten as $O(\sqrt{\frac{s}{t}}) = \varepsilon||\mathbf{P} - \mathbf{P}_k||_F^2$. Besides, $\sum_{i=k+1}^{s+k} \sigma_i^2(\mathbf{P}) \leq \sum_{i=k+1}^{n} \sigma_i^2(\mathbf{P}) = ||\mathbf{P} - \mathbf{P}_k||_F^2$. Hence, it can be obtained that

$$\mathrm{Tr}\left(\mathbf{\Pi}(\mathbf{K} - \widetilde{\mathbf{K}}_t)\mathbf{\Pi}\right) \leq (\varepsilon + \frac{k}{s})||\mathbf{P} - \mathbf{P}_k||_F^2.$$

$\square$

It can be obtained that

$$||(\mathbf{I}_n - \mathbf{\Pi})\mathbf{P}||_F^2 - ||(\mathbf{I}_n - \mathbf{\Pi})\widetilde{\mathbf{P}}_t||_F^2 = \text{Tr}((\mathbf{I}_n - \mathbf{\Pi})\mathbf{P}\mathbf{P}^\top) - \text{Tr}((\mathbf{I}_n - \mathbf{\Pi})\widetilde{\mathbf{P}}_t\widetilde{\mathbf{P}}_t^\top)$$
$$= \text{Tr}(\mathbf{P}\mathbf{P}^\top - \widetilde{\mathbf{P}}_t\widetilde{\mathbf{P}}_t^\top) - \text{Tr}(\mathbf{\Pi}(\mathbf{P}\mathbf{P}^\top - \widetilde{\mathbf{P}}_t\widetilde{\mathbf{P}}_t^\top)\mathbf{\Pi}).$$

Let $\alpha = \text{Tr}(\mathbf{P}\mathbf{P}^\top - \widetilde{\mathbf{P}}_t\widetilde{\mathbf{P}}_t^\top)$, and then the above equation can be rewritten as

$$||(\mathbf{I}_n - \mathbf{\Pi})\mathbf{P}||_F^2 + \text{Tr}(\mathbf{\Pi}(\mathbf{P}\mathbf{P}^\top - \widetilde{\mathbf{P}}_t\widetilde{\mathbf{P}}_t^\top)\mathbf{\Pi}) = \alpha + ||(\mathbf{I}_n - \mathbf{\Pi})\widetilde{\mathbf{P}}_t||_F^2.$$

After sufficient iterations, both $\alpha$ and $\text{Tr}(\mathbf{\Pi}(\mathbf{P}\mathbf{P}^\top - \widetilde{\mathbf{P}}_t\widetilde{\mathbf{P}}_t^\top)\mathbf{\Pi})$ are non-negative with a high probability. Thus, by Lemma 5 it holds that

$$||(\mathbf{I}_n - \mathbf{\Pi})\mathbf{P}||_F^2 \le \alpha + ||(\mathbf{I}_n - \mathbf{\Pi})\widetilde{\mathbf{P}}_t||_F^2$$
$$= ||(\mathbf{I}_n - \mathbf{\Pi})\mathbf{P}||_F^2 + \text{Tr}(\mathbf{\Pi}(\mathbf{P}\mathbf{P}^\top - \widetilde{\mathbf{P}}_t\widetilde{\mathbf{P}}_t^\top)\mathbf{\Pi}) \quad (15)$$
$$\le (1 + \varepsilon + \frac{k}{s})||(\mathbf{I}_n - \mathbf{\Pi})\mathbf{P}||_F^2.$$

Based on (15), Theorem 3 can be proved as follows. Let $\mathbf{\Pi} = \widetilde{\mathbf{Y}}_t\widetilde{\mathbf{L}}_t\widetilde{\mathbf{Y}}_t^\top$, where $\widetilde{\mathbf{Y}}_t$ is the indicator matrix obtained by applying a $\gamma$-approximate algorithm to $\widetilde{\mathbf{P}}_t$, then

$$||(\mathbf{I}_n - \widetilde{\mathbf{Y}}_t\widetilde{\mathbf{L}}_t\widetilde{\mathbf{Y}}_t^\top)\mathbf{P}||_F^2 \le \alpha + ||(\mathbf{I}_n - \widetilde{\mathbf{Y}}_t\widetilde{\mathbf{L}}_t\widetilde{\mathbf{Y}}_t^\top)\widetilde{\mathbf{P}}_t||_F^2$$
$$\le \alpha + \gamma||(\mathbf{I}_n - \widetilde{\mathbf{Y}}_t^*\widetilde{\mathbf{L}}_t^*\widetilde{\mathbf{Y}}_t^{*\top})\widetilde{\mathbf{P}}_t||_F^2,$$

where $\widetilde{\mathbf{Y}}_t^*$ is the optimal indicator matrix for the linear $k$-means problem on $\widetilde{\mathbf{P}}_t$. Since $\gamma > 1$, it follows that

$$\alpha + \gamma||(\mathbf{I}_n - \widetilde{\mathbf{Y}}_t^*\widetilde{\mathbf{L}}_t^*\widetilde{\mathbf{Y}}_t^{*\top})\widetilde{\mathbf{P}}_t||_F^2 \le \alpha + \gamma||(\mathbf{I}_n - \mathbf{Y}^*\mathbf{L}^*\mathbf{Y}^{*\top})\widetilde{\mathbf{P}}_t||_F^2$$
$$\le \gamma(1 + \varepsilon + \frac{k}{s})||(\mathbf{I}_n - \mathbf{Y}^*\mathbf{L}^*\mathbf{Y}^{*\top})\mathbf{P}||_F^2.$$

Thus,

$$||(\mathbf{I}_n - \widetilde{\mathbf{Y}}_t\widetilde{\mathbf{L}}_t\widetilde{\mathbf{Y}}_t^\top)\mathbf{P}||_F^2 \le \gamma(1 + \varepsilon + \frac{k}{s})||(\mathbf{I}_n - \mathbf{Y}^*\mathbf{L}^*\mathbf{Y}^{*\top})\mathbf{P}||_F^2,$$

which is equivalent to $f_K(\widetilde{\mathbf{Y}}_t) \le \gamma(1 + \varepsilon + \frac{k}{s}) \min_{\mathbf{Y}} f_K(\mathbf{Y})$.

# F  PRIVACY PRESERVATION PROPERTY OF FEDERATED KERNEL $k$-MEANS

## F.1  RECOVER USERS' DATA FROM RANDOM FEATURE MATRICES

A random feature for a data sample $\mathbf{x}_i$ has the form $\cos(\boldsymbol{\omega}^\top\mathbf{x}_i + b)$ where the $\boldsymbol{\omega}$ and $b$ are determined by the cloud server. Since the value of $\boldsymbol{\omega}^\top\mathbf{x}_i + b$ cannot be arbitrarily large, the number of its possible values is limited. If enough such random features are collected, the cloud server can determine the value of $\boldsymbol{\omega}^\top\mathbf{x}_i + b$ for each random feature, and then recover $\mathbf{x}_i$ by solving a system of linear equations.

## F.2  PROOF OF THEOREM 4

We then prove that the cloud server can at most recover the matrices $\{\mathbf{W}_t[m]^\top\mathbf{W}_t[m], m = 1, ..., M\}$ (only the multiplication $\mathbf{W}_t[m]^\top\mathbf{W}_t[m]$ not the matrix $\mathbf{W}_t[m]$) from the local computational results (e.g., $\mathbf{W}_t[m]^\top\mathbf{W}_t[m]\mathbf{c}_q$). The eigenpairs $\mathbf{W}_t[m]^\top\mathbf{W}_t[m]$ are determined via the distributed Lanczos algorithm. Since

$$\mathbf{W}_t[m]^\top\mathbf{W}_t[m] = \begin{bmatrix} \frac{\eta_t}{D}\mathbf{A}_t[m]^\top\mathbf{A}_t[m] & \sqrt{\frac{\eta_t(1-\eta_t)}{D}}\mathbf{A}_t[m]^\top\mathbf{B}_t[m] \\ \sqrt{\frac{\eta_t(1-\eta_t)}{D}}\mathbf{B}_t[m]^\top\mathbf{A}_t[m] & (1-\eta_t)\mathbf{B}_t[m]^\top\mathbf{B}_t[m] \end{bmatrix},$$

$\mathbf{A}_t[m]^\top\mathbf{A}_t[m]$ can be recovered from $\mathbf{W}_t[m]^\top\mathbf{W}_t[m]$.

For a matrix $\mathbf{A}_t[m] \in \mathbb{R}^{n_m \times D}$ ($n_m < D$), a matrix $\mathbf{A}' \in \mathbb{R}^{n_m \times D}$ can be constructed via

$$\mathbf{A}' = \mathbf{U}_o \mathbf{A}_t[m],$$

where $\mathbf{U}_o \in \mathbb{R}^{n_m \times n_m}$ is an arbitrary orthogonal matrix with $\mathbf{U}_o^\top \mathbf{U}_o = \mathbf{I}_n$. By this construction, it can be derived that

$$\mathbf{A}'^\top \mathbf{A}' = \mathbf{A}_t[m]^\top \mathbf{U}_o^\top \mathbf{U}_o \mathbf{A}_t[m] = \mathbf{A}_t[m]^\top \mathbf{I}_n \mathbf{A}_t[m] = \mathbf{A}_t[m]^\top \mathbf{A}_t[m].$$

Since there exist infinite matrices $\mathbf{U}_o$ satisfying $\mathbf{U}_o^\top \mathbf{U}_o = \mathbf{I}_n$, the problem $\mathbf{A}_t[m]^\top \mathbf{A}_t[m] = \mathbf{A}'^\top \mathbf{A}'$ has infinite solutions. Hence, recovering the random feature matrix $\mathbf{A}_t[m]$ from $\mathbf{A}_t[m]^\top \mathbf{A}_t[m]$ is an ill-posed problem with infinite solutions.

Since by employing CEM, the cloud cannot recover the random feature matrices via matrix operations, according to Section F.1, it is infeasible for the cloud server to recover users' data by solving a system of linear equations.

## G  ADDITIONAL EXPERIMENTAL SETTINGS

The three public datasets (Mushrooms, MNIST, Covtype)[3] are selected from the LIBSVM dataset repository. The Smartphone dataset is provided by a company. The smartphone dataset contains the power consumption data of one app on users' smartphones. Its twelve features represent the power consumption on twelve hardware components. The clustering task for this dataset is to find the distinct usage patterns of the app based on the power consumption data. For the concern of privacy, the Smartphone dataset will not be disclosed.

The description of the four existing methods used in the experiments are listed as follows.

1. Centralized kernel $k$-means Zha et al. (2001) (denoted as CK $k$-means): directly perform truncated SVD on the kernel matrix $\mathbf{K} = \mathbf{U}\boldsymbol{\Lambda}\mathbf{U}^\top$ to obtain a matrix that consists of the first $s$ column vectors of $\mathbf{U}\boldsymbol{\Lambda}^{\frac{1}{2}}$, and then apply linear $k$-means to this matrix;

2. Scalable kernel $k$-means Wang et al. (2019) (denoted as SK $k$-means): utilize the Nyström method to approximate the kernel matrix $\mathbf{K}$, and conduct kernel $k$-means over the approximated kernel matrix;

3. Distributed kernel $k$-means with random feature (Chitta et al., 2012) (denoted as RFK $k$-means): first transform the raw data samples to the corresponding random vector via the random Fourier feature method (Rahimi & Recht, 2008) and then utilize a distributed linear $k$-means to find the clusters in space of these random features;

4. Communication efficient distributed kernel PCA Balcan et al. (2016) (denoted as CE PCA): first conduct dimension reduction on the raw data samples through the communication efficient kernel PCA that integrates subspace embedding and adaptive sampling techniques to perform approximated kernel PCA in a distributed manner, and then apply a distributed linear $k$-means algorithm to the data samples after the dimension reduction.

For three distributed algorithms (FK $k$-means, RFK $k$-means, and CE PCA), the distributed linear $k$-means algorithm developed in (Balcan et al., 2013) is utilized to obtain the clustering results. Thus, the number of data samples $C$ in the coreset should be assigned.

For FK $k$-means, the maximal iteration number $T$ is selected from $[10, 20, 30, 40, 50]$. In the experiments of the Mushrooms dataset, $C$ is set to 1000. In the experiment of the MNIST dataset, $C$ is set to 1000. In the experiment of the Covtype dataset, $C$ is set to 4000. In the experiment of the Smartphone dataset, $C$ is set to 4000.

RFK $k$-means has two hyperparameters: the kernel parameter $\gamma$, and the number of random features $D$. The hyperparameter configuration for RFK $k$-means is set as follows. The value of $\gamma$ is the same as that of FK $k$-means. In the experiments of the Mushrooms dataset, $D = 200$, and $C$ is selected from $[100, 300, 500, 700]$. In the experiment of the MNIST dataset, $D = 800$, and $C$ is

---

[3]These datasets can be downloaded from
`https://www.csie.ntu.edu.tw/~cjlin/libsvmtools/datasets/`.

selected from $[100, 300, 500, 700, 900]$. In the experiment of the Covtype dataset, $D = 100$, and $C$ is selected from $[1000, 2000, 3000, 4000, 5000]$. In the experiment of the Smartphone dataset, $D = 50$, and $C$ is selected from $[500, 1000, 2000, 3000]$.

CE PCA has six hyperparameters: the kernel parameter $\gamma$, the number of principle components $d$ after PCA, the number of random features $D$, the subspace embedding dimension for the feature expansion $d_s$, The subspace embedding dimension for the data points $d_p$, and the number of representative points $p$. The hyperparameter configuration for CE PCA is set as follows. The value of $\gamma$ is the same as that of FK $k$-means In the experiments over different datasets, some hyperparameters of these two methods are not changed. For CE PCA, $d_s = 50$, $d_p = 250$, and $p = 500$. Besides, the number of principle components $d$ in CE PCA is the same as the number of eigenvectors $s$ in FK $k$-means. To obtain different communication cost and normalized mutual information scores, the value of $D$ is set to different values for both methods. In the experiment over the Mushrooms dataset, $D$ is selected from $[20, 50, 100, 200]$, and $C$ is set to 1000. In the experiment over the MNIST dataset, $D$ is selected from $[100, 200, 400, 800]$, $C$ is set to 1000. In the experiment over the Covtype dataset, $D$ are selected from $[20, 50, 100, 200]$, and $C$ is set to 4000. In the experiment over the Smartphone dataset, $D$ is selected from $[20, 50, 100]$, $C$ is set to 4000.

The discussion of the configuration of $D$ is as follows. For RFK $k$-means and CE PCA, $D$ is usually set to large values (more than 100). While for federated kernel $k$-means, $D$ can be set to relatively small values (less than 50). The reason is as follows. RFK $k$-means (and CE PCA) only employs random feature once to estimate the kernel matrix. Thus, it requires a large number of random features to obtain an estimation of the kernel matrix with low approximation error, and furthermore a high NMI score. In contrast, federated kernel $k$-means is an iterative algorithm where random features are employed in each iteration to reduce the gap between the estimation and the kernel matrix. Hence, the number of random feature is not necessary to be set to a large value in each iteration.

The communication cost of the three algorithms are determined as follows. For FK $k$-means, the communication cost is the communication cost of DSPGD with CEM plus the that of the distributed linear $k$-means. For RFK $k$-means, its communication cost equals the communication cost of the distributed linear $k$-means. For CE PCA, its communication cost is the communication cost of performing distributed PCA plus the communication cost of the distributed linear $k$-means. The communication cost the distributed linear $k$-means equals the number of data samples in the coreset times the dimension of a data sample. In both FK $k$-means and CE PCA, the dimension of a data sample equals the number of eigenvectors $s$. In RFK $k$-means, the dimension of a data sample equals the number of random features $D$ since the raw data cannot be exposed to the cloud server, and only the random feature vectors can be uploaded to the cloud server.

