# OpenReview forum: "A Communication Efficient Federated Kernel $k$-Means"
_ICLR.cc/2021/Conference — Reject_

### Official Review · AnonReviewer3 · 2020-10-28

**Rating:** 5
**Confidence:** 2

**Review:**

Summary:

This paper proposes a federated kernel k-means algorithm (FK k-means). The algorithm consists of two parts: a distributed stochastic proximal gradient descent (DSPGD) update rule, and a communication efficient mechanism (CEM) to reduce the communication cost. Instead of solving the original integer programming problem, the authors consider the relaxed convex stochastic composite optimization (SCO) problem and extends it to a distributed setting.

The main contribution is that the authors show, both theoretically and experimentally, that their proposed algorithm converges to the true solution of the SCO problem at O(1/T) rate. It is also proved that the communication cost does not grow with the number of samples N. In addition the authors characterize the error ratio between their algorithm and the original k-means. At the end the authors prove that the central server cannot recover the feature matrices. The authors compare their algorithm with other k-means algorithms on several datasets.


Pros:

- The federated setting of k-means with privacy preservation property is interesting and relatively new.

- The presentation of the theorems is very clear. The authors put interpretation after each statement and it is easy to follow most of the time. In particular it is shown that the proposed algorithm approaches the baseline scalable kernel k-means algorithm as T increases, both theoretically and experimentally.

- I appreciate the discussion of the motivation and related works in the first two section of the paper.


Cons:

- The whole Section 4 of the paper is filled with technical details and hard to follow. Considering the page limit, why not put the algorithms here, and put the details in the appendix? In addition the authors can consider putting important observations as lemmas.

- It seems the proof techniques are somehow standard. The whole proof of Theorem 1 feels like an extension to [Zhang et al., 2016] in a distributed setting. I did not check the other proofs though.

- My biggest concern is about novelty. Currently the proposed algorithm is heavily influenced by [Zhang et al., 2016] and [Wang et al., 2019]. If that is not the case, the authors can consider including a table to illustrate the difference between the algorithms.

------------------------------

Post-rebuttal:

I appreciate the authors' feedbacks. However the authors' response to the proof of Theorem 1 is not the most convincing, which is a big part of the claimed contribution.

---

> ### Author Response · Authors · 2020-11-24
> **Responses to Q1~Q3**
>
> We thank the reviewer for the positive comments and constructive suggestions on our paper.
>
> 1. The whole Section 4 of the paper is filled with technical details and hard to follow. Considering the page limit, why not put the algorithms here, and put the details in the appendix?
>
> We have moved the pseudo code of the algorithm back to Section 4.2, and moved some technical details to the appendix.
>
>
> 2. It seems the proof techniques are somehow standard. The whole proof of Theorem 1 feels like an extension to [Zhang et al., 2016] in a distributed setting.
>
> The difference between the proof in [Zhang et al., 2016] and our proof is as follows.
>   In [Zhang et al., 2016], the $(t+1)$-th solution $\mathrm{Z}_{t+1}^*$ determined by the stochastic proximal gradient descent (SPGD) is the exact solution to the following problem
>
> $$
> \min_{\mathrm{Z} \in \mathbb{R}^{n \times n}}{\frac{1}{2}||\mathrm{Z} - \mathrm{Z}_{t}||_F^2 + \eta_t \langle \mathrm{Z}}-\mathrm{Z}_t,  \mathrm{Z}_t - \mathrm{\xi}_t \rangle + \eta_t \lambda ||\mathrm{Z}||*
> $$
>
> However, in FK $k$-means, to reduce the communication cost, the $(t+1)$-th solution $\mathrm{Z}_{t+1}$ determined by the distributed stochastic proximal gradient descent (DSPGD) is an approximate solution.
> Since it is unknown how these approximate solutions affect the convergence of DSPGD, we need to prove that the approximate solutions do not affect the convergence of DSPGD. For the details of the proof, please refer to the proof of Theorem~1 (from the equation (6) to equation (15)).
>
>
>
> 3. My biggest concern is about novelty. Currently the proposed algorithm is heavily influenced by [Zhang et al., 2016] and [Wang et al., 2019]. If that is not the case, the authors can consider including a table to illustrate the difference between the algorithms.
>
> The novelties of FK $k$-means are as follows. First, we design DSPGD to approach the top-$s$ eigenpairs of the kernel matrix $\mathrm{K}$ under federated settings where raw data are maintained by users and the cloud cannot access the raw data. Second, a communication efficient mechanism (CEM) is designed to highly reduce the communication cost of DSPGD.
>
> Among these two novelties, DSPGD is a counterpart of the stochastic proximal gradient descent (SPGD) algorithm in stochastic PCA [Zhang et al., 2016]. However, DSPGD is distinct from SPGD in the following three features.
>
> DSPGD is conducted under federated settings. However, SPGD is conducted in a centralized manner where users' raw data are collected at the cloud server.
>
> The communication cost is considered in the design of DSPGD, which results in CEM, while the communication cost is not considered in SPGD.
>
> Although both DSPGD and SPGD aim at approaching the top eigenpairs of $\mathrm{K}$, in the t-th iteration, DSPGD only needs to obtain an approximate solution $\mathrm{Z}\_{t+1}$ instead of the exact solution $\mathrm{Z}_{t+1}^*$ to the same problem like SPGD, which leads to less communication cost under federated settings.
>
> oreover, FK $k$-means is distinct from the scalable kernel $k$-means [Wang et al., 2019] (denoted as SK $k$-means) in the following two features.
>
> FK $k$-means approaches the top eigenvectors without collecting users' raw data or users' random features at the cloud server. However, SK $k$-means requires users to upload some of their raw data or random features to the cloud server to approximate the kernel matrix $\mathrm{K}$.
>
> FK $k$-means is an iterative algorithm, and it can approach the top eigenvectors of $\mathrm{K}$ more accurately by employing more iterations. In contrast to FK $k$-means, SK $k$-means is a one-shot algorithm, i.e., it only determines an approximate kernel matrix $\tilde{\mathrm{K}}$ once and then applies SVD to $\tilde{\mathrm{K}}$ to approach the top eigenvectors of $\mathrm{K}$. Thus, the accuracy of these top eigenvectors is limited by the number of data samples or random features uploaded by users.
>
> Reference
>
> [Zhang et al., 2016] Lijun Zhang, Tianbao Yang, Jinfeng Yi, Rong Jin, and Zhi-Hua Zhou. Stochastic optimization for kernel PCA. In Proceedings of the 30th AAAI Conference on Artificial Intelligence, pp. 2316–
> 2322, 2016.
>
> [Wang et al., 2019] Shusen Wang, Alex Gittens, and Michael W Mahoney. Scalable kernel k-means clustering with Nystr{\"o}m approximation: relative-error bounds. The Journal of Machine Learning Research, 20
> (1):431–479, 2019

---

### Official Review · AnonReviewer4 · 2020-10-28
**The paper is not novel by itself but is technically solid and proposed a new distributed clustering algorithm.**

**Rating:** 5
**Confidence:** 3

**Review:**

The paper proposed a new distributed kernel k-means algorithm that has lower communication overhead compared to the available methods while does not require transmitting the data samples from agents to the master node and therefore claims to preserve the privacy of agents. The paper is rather difficult to follow and the novelty of the paper is not very clear to me. It seems to me that the major contribution of the paper is to come up with efficient tricks during the implementation of the distributed Lanczos algorithm (DLA) to find the eigenvalues of the kernel approximated by using the well-known random Fourier features (Rahimi, nips 2008). So the end result is not something utterly novel but rather an efficient way of utilizing available tools to design a new distributed algorithm.

There are several points that I think need to be clarified:
1. The literature review does not clearly illustrate the contribution of the paper. For instance, in Section 2.1, the author says "However, these algorithms are designed with an assumption that they are executed at the cloud server where the constraint on transmissions of the raw data samples does not exist." Does this mean that your work is the first to constrain the transmission of data samples? If so you should clearly say so. Besides, why transmitting the raw data sample is important here? Is it due to privacy issues? Because the way I see it, one does not need to transmit the raw data samples but rather its local kernel matrix K, and doing so does not necessarily endanger the privacy of an agent, since one cannot easily recover data samples from K.

2. In Section 6.2, the author says "According to the results in the four subfigures, it is shown that CEM can reduce communication cost of DSPGD by more than 95%." It would be interesting to explain what values of r_t, Q_0, and Q_1 leads to this improvement and why.

3. The 60% communication cost reduction mentioned throughout the paper seems a bit exaggerated as it does not consistently happen. It only happens for the MNIST dataset in Figure 3(b). Do you know why? My guess is that due to the high sparsity level of MNIST samples, the kernel matrix might contain a lot of zeros leading to such behavior.

4. What about the drawbacks of the proposed method? What I see is a very complicated algorithm that requires heavy computational resources at each agent, which makes it unsuitable for the toy example explained in the introduction about smartphones.

After rebuttal: I thank the author for their response but I have to lower my rating by one step after reading the comments of Reviewer2.

---

> ### Author Response · Authors · 2020-11-24
> **Response to summary and Q1**
>
> We thank the reviewer for the constructive suggestions on our paper.
>
> Summary: The paper is rather difficult to follow and the novelty of the paper is not very clear to me. It seems to me that the major contribution of the paper is to come up with efficient tricks during the implementation of the distributed Lanczos algorithm (DLA) to find the eigenvalues of the kernel approximated by using the well-known random Fourier features (Rahimi, nips 2008).
>
> In Section 4.1, we first point out that the key problem for designing FK $k$-means is to obtain the top eigenpair of the kernel matrix $\mathrm{K}$ in a distributed manner. To solve this problem, we design a distributed stochastic proximal gradient descent (DSPGD) algorithm. We then present the challenges on solving the key problem and how we resolve these challenges.
>  Since $\mathrm{K}$ is not available under federated settings. To this end, an estimate of $\mathrm{K}$, denoted as $\mathrm{\xi}$, is constructed jointly at users' devices based on random features [1] of local data samples.
>   Since $\mathrm{\xi}$ is distributed among different devices, it is processed by the distributed Lanczos algorithm (DLA) [3] to obtain an estimate of $\mathrm{K}$, i.e., $\mathrm{Z}$, at the cloud server. Afterwards, an approximate version of the top eigenpairs of $\mathrm{K}$ can be obtained from $\mathrm{Z}$ through singular value decomposition (SVD).
>  To improve the accuracy of approximation, $\mathrm{Z}$ is iteratively updated by DSPGD.
>
>   More specifically, in the $t$-th iteration, an estimate $\mathrm{\xi}\_{t}$ is constructed at users' devices, and then the estimate $\mathrm{Z}\_{t}$ at the cloud server is updated to $\mathrm{Z}\_{t+1}$ by applying DLA to a weighted sum of $\mathrm{Z}\_{t}$ and $\mathrm{\xi}\_{t}$.
>
>   In Section 4.2, we first point out that the process of obtaining an updated $\mathrm{Z}_{t}$ results in high communication cost, because DLA is applied to matrices with the number of rows/columns equals the number of data samples. To this end, we design a communication efficient mechanism (CEM) so that DLA is applied to a new type of matrices. We then present the design of the new type of matrices and the procedure of obtaining the top eigenpairs of $\mathrm{K}$ in CEM.
>
>   The major novelty of our work is that we design a new distributed stochastic proximal gradient descent (DSPGD) algorithm to approximate the top-$s$ eigenpairs of the kernel matrix $\mathrm{K}$ under the federated settings where raw data are maintained by users and the cloud cannot access the raw data. Besides, we design a communication efficient mechanism (CEM) that can highly reduce the communication cost of DSPGD. In addition, we derive the convergence rate of DSPGD, analyze the communication cost of DSPGD before and after employing CEM, and analyze the clustering quality of FK $k$-means.
>
> 1. The literature review does not clearly illustrate the contribution of the paper. For instance, in Section 2.1, the author says ``However, these algorithms are designed with an assumption that they are executed at the cloud server where the constraint on transmissions of the raw data samples does not exist.'' Does this mean that your work is the first to constrain the transmission of data samples? If so you should clearly say so.
>
> We have revised the literature review and clearly pointed out the differences between our work and the existing schemes.
>
> In Section 2.1, the clause ``where the constraint on transmissions of the raw data samples does not exist'' is indeed confusing. We actually want to express that the distributed kernel $k$-means schemes mentioned in Section 2 are designed for the cloud server where users' raw data are collected. In contrast to these distributed kernel $k$-means schemes, FK $k$-means is the first scheme to conduct the kernel $k$-means clustering without uploading users' raw data to the cloud server. Besides, FK $k$-means also considers the communication efficiency during the clustering task.

---

> > ### Author Response · Authors · 2020-11-24
> > **Response to Q2 ~Q3**
> >
> > 2. Besides, why transmitting the raw data sample is important here? Is it due to privacy issues? Because the way I see it, one does not need to transmit the raw data samples but rather its local kernel matrix $\mathrm{K}_m$, and doing so does not necessarily endanger the privacy of an agent, since one cannot easily recover data samples from $\mathrm{K}_m$.
> >
> > Not uploading the raw data to the cloud server is important for two reasons. First, the transmission of the raw data can consume large communication bandwidth. Second, uploading the raw data can lead to privacy issues.
> >
> >   If each user $m$ uploads its local kernel matrix $\mathrm{K}_m$ to the cloud server, indeed it is difficult for the cloud server to recover users' data samples from $\mathrm{K}_m$. However, it is non-trivial to conduct the kernel $k$-means clustering using $\{ \mathrm{K}_m, m=1,...,M\}$. Based on $\{ \mathrm{K}_m, m=1,...,M\}$, the cloud server can construct a block-diagonal matrix $\tilde{\mathrm{K}}$.
> >
> >
> > Note that $\tilde{\mathrm{K}}$ is far from the true kernel matrix $\mathrm{K}$. To the best of our knowledge, none of existing schemes can accurately approach the top eigenvectors of $\mathrm{K}$ based on $\tilde{\mathrm{K}}$. Besides, uploading $\mathrm{K}_m$ can lead to even larger communication cost than uploading users' raw data. Assume each user $m$ has $n_m$ $d$-dimension data samples, and each dimension is represented by a $4$-byte floating-point number. Thus, the communication cost of uploading the raw data is $4 n_m d$ bytes for each user $m$.
> >   Since $\mathrm{K}_m$ is a symmetric matrix, the upper triangular portion of $\mathrm{K}_m$ actually contains all the information of $\mathrm{K}_m$. Thus, the communication cost of uploading $\mathrm{K}_m$ is $2 n_m (n_m + 1) $ bytes. If $n_m > 2d-1$, which is common for a dataset, the communication cost of uploading $\mathrm{K}_m$ is larger than that of uploading the raw data. Thus, it is not communication efficient to upload $\mathrm{K}_m$ to the cloud server.
> >
> > 3. In Section 6.2, the author says ``According to the results in the four subfigures, it is shown that CEM can reduce communication cost of DSPGD by more than $95$%.'' It would be interesting to explain what values of $r_t$, $Q_0$, and $Q_1$ leads to this improvement and why.
> >
> > The communication ratio (i.e., the ratio of the communication cost of DSPGD without CEM to that of DSPGD with CEM) equals $\frac{(N+MD)Q_0}{M(r_t+D)Q_1}$ according to Theorem 2. In this expression, the ratio $\frac{Q_0}{Q_1} \approx 1$, which can be seen in Figure 4 that is newly added to the appendix. As a result, the communication ratio mainly relies on $\frac{N+MD}{M(r_t+D)}$.
> >
> >   We set $r_t$ to its upper bound to compute a lower bound of $\frac{N+MD}{M(r_t+D)}$. According to Figure 5 in the appendix, the value of $r_t$ is set to $12$, $20$, $22$, $10$ for the Mushroom dataset, the MNIST dataset, the Covtype dataset, the Smartphone dataset, respectively. For other variables $\frac{N+MD}{M(r_t+D)}$, the values of $N$ and $D$ are given in Table 1, and $M=5$. As a result, the lower bound of $\frac{N+MD}{M(r_t+D)}$ equals $60.73$, $55.45$, $2235.23$, $1142.76$ for the Mushroom dataset, the MNIST dataset, the Covtype dataset, the Smartphone dataset, respectively. The statement ``CEM can reduce communication cost of DSPGD by more than $95$%'' is equivalent to the communication ratio larger than $20$. Since the lower bound of $\frac{N+MD}{M(r_t+D)}$ is larger than $20$ for each dataset, it is concluded that CEM can reduce communication cost of DSPGD by more than $95$%.
> >
> >   Note that the statement ``more than $95\%$'' is conservative. According to the lower bound of $\frac{N+MD}{M(r_t+D)}$, CEM can actually reduce the communication cost of DSPGD by more than $98$%, which is also consistent with the results in Figure 2. We have revised the corresponding statements in the revised paper.

---

> > > ### Author Response · Authors · 2020-11-24
> > > **Response to Q4 and Q5**
> > >
> > > 4. The $60$% communication cost reduction mentioned throughout the paper seems a bit exaggerated as it does not consistently happen. It only happens for the MNIST dataset in Figure 3(b). Do you know why? My guess is that due to the high sparsity level of MNIST samples, the kernel matrix might contain a lot of zeros leading to such behavior.
> > >
> > > Indeed the $60$% communication cost reduction does not hold for the Covtype dataset.
> > >   However, we would like to point out that the $60\%$ communication cost reduction not only holds for the MNIST dataset, but also holds for the Mushrooms dataset and the Smartphone dataset, which can be seen from Figure 3(a) and 3(d).
> > >
> > >   The reasons for the $60$% communication cost reduction are explained as follows. First, for each of the three datasets, the sparsity of the data samples leads to many zero items in its kernel matrix $\mathrm{K}$. As a result, the remaining non-zero items in $\mathrm{K}$ can be approached with only a small $D$ and a small number of iterations of DSPGD. Second, among the top eigenvalues of $\mathrm{K}$, the gap between two adjacent eigenvalues (e.g., the $i$-th eigenvalues and the $i+1$-th eigenvalues) is large. As a result, these top eigenvalues are approached with only a small number of Lanczos iterations, which reduces the communication cost in each iteration of DSPGD.
> > >
> > > 5. What about the drawbacks of the proposed method? What I see is a very complicated algorithm that requires heavy computational resources at each agent, which makes it unsuitable for the toy example explained in the introduction about smartphones.
> > >
> > > The drawbacks of FK $k$-means mainly come from two aspects. First, FK $k$-means is a synchronous algorithm. The efficiency of the algorithm can be affected by stragglers. Second, FK $k$-means requires all the users to participate the whole training process. If one user drops out halfway, the algorithm may have to be restarted.
> > >
> > >   In FK $k$-means (we consider the communication efficient mechanism (CEM) here), the algorithm conducted at each user's device has low computational cost, and it is easy for the common smartphone CPUs with tens of GFLOPS to execute the algorithm. The computational cost is analyzed as follows. In the $t$-th iteration of DSPGD with CEM, the $m$-th user with $n_m$ data samples needs to compute ${\mathrm{g}}_q = {\mathrm{W}}_t[m]^\top {\mathrm{W}}_t[m] {\mathrm{c}}_q$ for $Q_t$ iterations. ${\mathrm{W}}_t[m]^\top {\mathrm{W}}_t[m]$ can be only computed once with $(2 n_m -1)(r_t + D)^2$ floating-point operations. Computing one ${\mathrm{g}}_q$ requires $2(r_t + D)^2 - (r_t + D)$ floating-point operations. Thus, one iteration of DSPGD with CEM requires $(2 n_m + 2 Q_t -1)(r_t + D)^2 - Q_t (r_t + D)$ floating-point operations at the $m$-th user's device. Among the four datasets, the MNIST dataset requires the largest communication cost. We then computes an upper bound of its communication cost by setting $n_m$, $r_t$, $D$, and $Q_t$ to their maximal values or upper bound values, i.e., $n_m=12000$, $r_t=20$, $D=200$, and $Q_t=26$. As a result, the computational cost of one iteration of DSPGD with CEM approximates $1$ giga floating-point operations that can be completed by common smartphone CPUs within tens of milliseconds.
> > >
> > > Reference
> > >
> > > [1] Brendan McMahan, Eider Moore, Daniel Ramage, Seth Hampson, and Blaise Aguera y Arcas.
> > > Communication-efficient learning of deep networks from decentralized data. In Proceedings of
> > > the 20th International Conference on Artificial Intelligence and Statistics, pp. 1273–1282, 2017.

---

### Official Review · AnonReviewer2 · 2020-10-28
**Not better than baseline; main theorem is wrong**

**Rating:** 1
**Confidence:** 5

**Review:**

##########################################################################

Summary:

This paper proposes a new method and two algorithms for solving kernel k-means. The contribution is that the algorithms converge to the optimal solution. The downsides are 1) the method is not better than a simple baseline (i.e., random features + distributed power method) and 2) the main theorem (Theorem 3) is wrong.


##########################################################################

Reasons for score:

I vote for rejection for two reasons. First, the proposed method appears not useful. The same problem can be solved in a much simpler and faster way. Second, the main theorem, Theorem 3, is wrong.


##########################################################################

Pros:

+ This paper develops a new method of distributed kernel k-means. The method is new, although I do not find it very useful.

+ This paper proves that two algorithms can correctly solve the trace-norm regularized problem in Section 4.1.


##########################################################################

Cons:

1. First and foremost, I do not see a good reason for using the proposed algorithm. The goal of the algorithm is to find the top singular vectors of the random features, $A$.

    - The solution to the trace norm regularized problem, $Z^*$, has the same singular vectors as K. By finding $Z^*$, you can find the eigenvectors of $K$.

    - However, the same goal could be achieved in an easier and less expensive way, i.e., random features + distributed power method. Random features are naturally distributed among the clients. Their truncated SVD could be found by the distributed power method or Krylov subspace methods. Truncated SVD is easier than solved the proposed trace norm regularized problem because the latter uses SVS which repeatedly performs SVD.

2. I am very surprised that Theorem 3 does not reply on $D$ (the number of random features). So I checked some of the proofs. I found Theorem 3, which is the main theorem, is wrong.

    - $Z^*$ has the same top eigenvectors as $\xi$. But $Z^*$ may not have the same as $K$.

    - The proof of Theorem 3 relies on that $Z^*$ has the same eigenvectors as $K$. This is wrong.


3. The description of the algorithm is difficult to follow. I’d suggest splitting algorithm description into 3 paragraphs: 1) Client-side computation, 2) server-side computation, and 3) communications.





Typos:

17th page: "The following two lemmas will be used in the proof of Theorem 4.” Do you mean Theorem 3?

##########################################################################

Updates after discussing with the authors

1. The paper is not very clearly written, and I had misunderstandings. Some of my comments above are not right.

2. However, I will not change my rating. I found the convergence rates stated in the paper are misleading. The paper claims $O(1/T)$ convergence rate. In fact, this is WRONG. The authors assume the Frobenius and trace norms of $n\times n$ matrices are CONSTANTS. This is not possible. The norms are $O(n)$. Simple arguments can show $|| \xi ||_F = G$ is $O(n)$.

3. Based on the right assumption that $|| \xi ||_F = G = O(n)$, the required number of iterations is $T = O(n^2)$. The algorithm is not communication-efficient. It is more expensive than communicating the $n\times n$ kernel matrices.

4. After reading my comments, the authors changed their notation from $G$ to $\gamma$, $C$, $G$, and $H$. They are also Frobenius and trace norms of $n\times n$ matrices. The authors assume $\gamma$, $C$, $G$ and $H$ are constants. This is WRONG. They are $O(n)$.

    - For example, if they use the bound of Rahimi and Recht,  then $|| \xi - K ||_F^2 = G^2$ is $O(n^2)$.  A bound as good as $|| \xi - K ||_F^2 = O(n)$ would surprise me; if the authors know such a bound, please let me know.

    - Let me strengthen my point again: IT IS WRONG TO ASSUME MATRIX NORMS ARE CONSTANTS! If the authors can prove they are constants, they need to show me the proofs. If they cannot, they should assume Frobenius norm and trace norm are $O(n)$.

---

> ### Author Response · Authors · 2020-11-24
> **Response to summary and Q1**
>
> We thank the reviewer for the constructive suggestions on our paper.
>
> Summary: This paper proposes a new method and two algorithms for solving kernel k-means. The contribution is that the algorithms converge to the optimal solution. The downsides are 1) the method is not better than a simple baseline (i.e., random features + distributed power method) and 2) the main theorem (Theorem 3) is wrong.
>
> The contribution of our work is as follows.
>   We design a new distributed stochastic proximal gradient descent (DSPGD) algorithm to approximate the top-$s$ eigenpairs of the kernel matrix $\mathrm{K}$ under federated settings where raw data are maintained by users and the cloud cannot access the raw data. Besides, we design a communication efficient mechanism (CEM) that can highly reduce the communication cost of DSPGD. In addition, we theoretically analyze the performance of FK $k$-means and its components in three aspects: the convergence rate of DSPGD; the communication cost of DSPGD before and after employing CEM; the clustering quality of FK $k$-means.
>
>   We would like to point out that our method (i.e., distributed stochastic proximal gradient descent (DSPGD)) is actually better than the baseline (i.e., random features + distributed power method). The baseline is actually a special case of the DSPGD. If DSPGD adopts the same value of $D$ as that in the baseline and is only executed for one iteration, then DSPGD is reduced to the baseline. For the comparison between DSPGD and the baseline, please refer to the response to comment 2.
>
>   Besides, we have checked the proof of Theorem 3 and confirmed that Theorem 3 is correct. For the detailed clarification, please refer to the responses to comment 3 and comment 4.
>
> 1. The goal of the algorithm is to find the top singular vectors of the random features $\mathrm{A}$. However, the same goal could be achieved in an easier and less expensive way, i.e., random features + distributed power method. Random features are naturally distributed among the clients. Their truncated SVD could be found by the distributed power method or Krylov subspace methods. Truncated SVD is easier than solved the proposed trace norm regularized problem because the latter uses SVS which repeatedly performs SVD.
>
> The goal of the distributed stochastic proximal gradient descent (DSPGD) is actually to approach the top eigenpairs of the kernel matrix $\mathrm{K}$, not the the top singular vectors and the corresponding singular values of the random features $\mathrm{A}$.
>   In FK $k$-means, DSPGD is designed to solve a stochastic composite optimization (SCO) problem in a distributed manner:
> $$
> \min_{\mathrm{Z} \in \mathbb{R}^{n \times n}}{\frac{1}{2}\mathbb{E}[||\mathrm{Z} - \mathrm{\xi}||_F^2]+\lambda ||\mathrm{Z}||*}.
> $$
>
> The optimal solution $\mathrm{Z}^*$ to SCO problem is
> $$
> \mathrm{Z}^* = \sum_{i:\lambda_i > \lambda}{(\lambda_i - \lambda) \mathrm{u}_i \mathrm{u}_i^\top},
> $$
>
> where ${\mathrm{u}}_i$ and $\lambda_i$ are the $i$-th eigenvector and the $i$-th eigenvalue of $\mathrm{K}$, respectively. Assume that the rank of $\mathrm{Z}^*$ equals $s$.
>   By applying SVD to $\mathrm{Z}^*$, its eigenvectors $\{{\mathrm{u}}_i, i=1,...,s\}$ of $\mathrm{K}$ and eigenvalues $\{\lambda_i - \lambda, i=1,...,s\}$ are determined. The eigenvectors of $\mathrm{Z}^*$ are exactly the top-$s$ eigenvectors of $\mathrm{K}$. Moreover, by adding $\lambda$ to the eigenvalues of $\mathrm{Z}^*$, the top-$s$ eigenvalues of $\mathrm{K}$ are obtained.
>   Thus, the top eigenpairs of $\mathrm{K}$ are approached by DSPGD.
>
> The baseline can indeed be used to approximate the top eigenvectors of $\mathrm{K}$. However, the number of random features $D$ should be set to a very large value in order to obtain an accurate estimate of $\mathrm{K}$ according to [1]. This method is feasible only if users' devices have enough memory to maintain these random feature vectors. Compared with the baseline, DSPGD is a generic method that provides a tradeoff between memory and computational time: the larger is $D$, the fewer iterations are required (an extreme case is that only one iteration is required) to approach the top eigenvectors of $\mathrm{K}$. Thus, DSPGD can be adapted to devices with different memory space.

---

> > ### Author Response · Authors · 2020-11-24
> > **Response to Q2~Q5**
> >
> > 2. I am very surprised that Theorem 3 does not rely on $D$ (the number of random features).
> >
> > $D$ actually affects the convergence rate of DSPGD, and furthermore affects the approximation ratio in Theorem 3, which is explained as follows. $D$ affects the variance of $\mathrm{\xi}_t$ according to [1]: the larger is $D$, the smaller is the variance of $\mathrm{\xi}_t$. Thus, a larger $D$ leads to a smaller $G$ for the inequality $||\mathrm{\xi}_t - \mathrm{K}||_F \leq G$ in Lemma 3.
> >
> > According to the following inequality in Lemma 3
> >  $$
> > ||\mathrm{Z}_{T+1} - \mathrm{Z}^*||_F^2 \leq \frac{4}{T}\left( C^2 + \lambda \gamma + 2G^2\tau + \frac{2}{3}GH\tau +GH\right),
> > $$
> >
> > we can see that a smaller $G$ leads to $\mathrm{Z}\_{T+1}$ closer to the optimal solution $\mathrm{Z}^*$, which means that $\mathrm{Z}_{T+1}$ converges faster to $\mathrm{Z}^*$. The inequality is then used in Lemma 5 to compute an upper bound of $\varepsilon$ in the approximate ratio (i.e., $1+\varepsilon+\frac{k}{s}$). According to Lemma 5, given $T$, the smaller is the bound, the smaller is $\varepsilon$, which concludes that $D$ affects the approximation ratio in Theorem 3
> >
> > However,  we can also see that different values of $D$ do not change the asymptotic convergence rate of DSPGD, i.e., $O(1/T)$. As a result, different values of $D$ also do not change the asymptotic convergence rate of $\varepsilon$, i.e., $O(\sqrt{s/T})$.
> >
> > 3. $\mathrm{Z}^*$ has the same top eigenvectors as $\mathrm{\xi}$. But $\mathrm{Z}^*$ may not have the same as $\mathrm{K}$. The proof of Theorem 3 relies on that $\mathrm{Z}^*$ has the same eigenvectors as $\mathrm{K}$.
> >
> > We would like to clarify that $\mathrm{Z}^*$ actually has the same top eigenvectors as $\mathrm{K}$ rather than as $\mathrm{\xi}$. Since $\mathrm{Z}^*$ is the optimal solution to SCO problem, it can be proved that as long as $\lambda$ is set to a proper value, $\mathrm{Z}^*$ has the same top eigenvectors as $\mathrm{K}$. However, the random matrix $\mathrm{\xi}_t$ is constructed by a random feature method in the $t$-th iteration of DSPGD (in DSPGD, $\mathrm{\xi}$ is replaced by $\mathrm{\xi}_t$). The top eigenvectors of $\mathrm{\xi}_t$ is unrelated to that of $\mathrm{Z}^*$.
> >
> > 4. The description of the algorithm is difficult to follow. I’d suggest splitting algorithm description into 3 paragraphs: 1) Client-side computation, 2) server-side computation, and 3) communications.
> >
> > To improve the clarity of algorithm description, we have marked each step with a number and pointed out the conductor (the cloud server or the users' devices) for each step. Please refer to the last paragraphs in Section 4.1 and Section 4.2.
> >
> > 5. 17th page: ``The following two lemmas will be used in the proof of Theorem 4.'' Do you mean Theorem 3?
> >
> > Indeed here Theorem 4 should be replaced with Theorem 3. We have corrected this typo in our revised version of paper.
> >
> > Reference
> >
> > [1] Ali Rahimi and Benjamin Recht. Random features for large-scale kernel machines. In Advances in
> > Neural Information Processing Systems 20, pp. 1177–1184, 2008.

---

> > ### Comment · AnonReviewer2 · 2020-11-25
> > **The actual convergence rate appears very different from the claimed**
> >
> > Thanks for the clarifications. I admit that I had some misunderstandings. I examined the pointed theorems and the proofs more carefully and found more problems.
> >
> > 1. Assume the statements in the authors' response are correct. Theorem 3 is not correctly stated. Theorem 3 does not say it depends on any assumption other than Def 1. I suppose the theorem assumes $E[\xi] = K$ and $|| \xi ||_F < L$, etc. Such assumptions must not be hidden. If you made the assumptions explicit, I would not have the misunderstandings.
> >
> > 2. Theorem 1 and Theorem 3 implicitly assume $|| \xi ||_F < L$ is a constant. This is wrong.
> > $\xi$ has $n^2$ elements; the elements are unbiased estimates of kernel functions which are constants. Thus $|| \xi ||_F^2 = O(n^2)$. $L$ is $O(n)$, not a constant.
> >
> > 3. Because $L=O(n)$, the convergence rate in Theorem 1 is not $O(1/T)$. Instead, the rate is $O(n^2 / T)$. It means you have to set $T = O(n^2)$. Then the overall communication complexity is worse than sending $n\times n$ kernel matrices. I am wondering if the algorithm is useful.
> >
> > 4. The same problem is with Theorem 3. Setting $T$ to $O(s)$ is not enough. You need to set $T$ to $O(s n^2)$.
> >
> > 5. There are still many typos in the proofs. You need to carefully proofread the proofs. Otherwise, I could not assume the proofs are correct.

---

> > > ### Author Response · Authors · 2020-11-25
> > > **Clarification to the convergence rate and responses to Q1~Q5**
> > >
> > > Thanks a lot for your constructive feedback.
> > >
> > > We agree that the assumption that $||\mathrm{\xi}||_F \leq L$ is not appropriate. In fact, we can eliminate this assumption without affecting the conclusions of Theorems 1 and 3, as explained below.
> > >
> > > After removing this assumption, the result of Theorem 1 becomes
> > > $$
> > > ||\mathrm{Z}\_{T+1} - \mathrm{Z}\_{T}||\_F^2 \leq \frac{4}{T}{\left( C^2 + \lambda \gamma + 2G^2\tau + \frac{2}{3}GH\tau +GH\right)},
> > > $$
> > >
> > > where $\gamma = \max\_{t \in [T]}{||\mathrm{Z}\_t||\_\star}$, $C^2=\max\_{t\in[T]}{||\mathrm{Z}\_{t} - \mathrm{\xi}\_{t}||\_F^2}$,  and assume that $||\mathrm{\xi}\_{t} - \mathrm{K}||\_F \leq G$ and $||\mathrm{Z}\_t - \mathrm{Z}^\star||\_F \leq H$ for all $t>2$.
> > >
> > > From this result we can see that given $T$, $C$, $\gamma$, $G$, and $H$ are all constants, so during the $T$ iterations the asymptotic convergence rate of DSPGD is still $O(1/T)$, and thus the result in Theorem 3 also holds.
> > >
> > > Here are the responses to the new comments
> > >
> > > 1.Assume the statements in the authors' response are correct. Theorem 3 is not correctly stated. Theorem 3 does not say it depends on any assumption other than Def 1. I suppose the theorem assumes $E[\xi] = K$ and $||\xi||_F < L$, etc. Such assumptions must not be hidden. If you made the assumptions explicit, I would not have the misunderstandings.
> > >
> > > For Theorem 1 and Theorem 3, we have added the assumptions used in their proofs to the statements of the corresponding theorems.
> > >
> > > 2.Theorem 1 and Theorem 3 implicitly assume $||\xi||_F < L$ is a constant. This is wrong. $\xi$ has $n^2$
> > >  elements; the elements are unbiased estimates of kernel functions which are constants. Thus $||\xi||_F^2 = O(n^2)$. $L$ is $O(n)$, not a constant.
> > >
> > > We agree with this comment so we have removed the assumption $||\xi||_F < L$ from Theorem 1 and revised the result of Theorem 1.
> > >
> > > 3.Because $L=O(n)$, the convergence rate in Theorem 1 is not $O(1/T)$. Instead, the rate is $O(n^2/T)$
> > > . It means you have to set $T=O(n^2)$. Then the overall communication complexity is worse than sending $n \times n$ kernel matrices. I am wondering if the algorithm is useful.
> > >
> > > According to the new result in Theorem 1, we can see that the convergence rate of DSPGD is still $O(1/T)$. Thus, we do not need to set $T=O(n^2)$ for the convergence of DSPGD
> > >
> > > 4.The same problem is with Theorem 3. Setting $T$ to $O(s)$ is not enough. You need to set $T$ to $O(sn^2)$.
> > >
> > > Since convergence rate of DSPGD is still $O(1/T)$, setting $T$ to $O(s)$ is enough for FK $k$-means
> > >
> > > 5.There are still many typos in the proofs. You need to carefully proofread the proofs. Otherwise, I could not assume the proofs are correct.
> > >
> > > We have proofread our paper and revised these typos.

---

### Official Review · AnonReviewer1 · 2020-10-29
**A distributed method for kernel k means with some performance guarantees**

**Rating:** 6
**Confidence:** 3

**Review:**

Summary:

This paper proposes a distributed version of kernel k-means clustering where some federated structure is used to do distributed processing on the data. Privacy and communication issues are also studied. Numerical results are provided.

Reasons for the score:

This paper seems provide a new algorithm for distributed clustering. However, the way the algorithm is presented look like a patch of a number of things coming together one after the other with no general structure. This might be caused by the fact that the algorithm is only presented in the appendix.

The paper is written in a convoluted manner. This is the main limitation, at some point, we are talking about k means, SVS, DLA,DSPGD, EVD, SPGD, a bunch of other methods that are coupled together towards the main approach.

Problem 1 seems to be an integer programming problem, thus with very high computation complexity. It is not clear how this is solved.

In the abstract please let me know what are those two levels of privacy you are talking about.

In the abstract, what does it mean that the clustering loss of the distributed method approaches the centralized one, please elaborate.

Why developing a federated learning algorithm is a promising approach? Please elaborate.

The second part of the intro turns into a detail technical analysis of the algorithm components, and so far we haven’t seen the algorithm so it all remains a technical abstract  discussion that takes away the main messages.

The algorithm is in the appendix, so the description and analysis is made on an item that has not been presented in the main text.

The way the result is presented makes it look like the proposed method is a concatenation of other results, rather than the solution of a technical challenge in the problem.

Numerical results are well presented,

---

> ### Author Response · Authors · 2020-11-24
> **Response to Q1 and Q2**
>
> We thank the reviewer for the constructive suggestions on our paper.
>
> 1. This paper seems provide a new algorithm for distributed clustering. However, the way the algorithm is presented look like a patch of a number of things coming together one after the other with no general structure. This might be caused by the fact that the algorithm is only presented in the appendix.
>
> To improve the presentation of the algorithm, we have revised the paper as follows. We have added an overview of federated kernel $k$-means (denoted as FK $k$-means) at the beginning of Section 4.
>
>   In Section 4.1, we first point out that the key problem for designing FK $k$-means is to obtain the top eigenpair of the kernel matrix $\mathrm{K}$ in a distributed manner. To solve this problem, we design a distributed stochastic proximal gradient descent (DSPGD) algorithm. We then present two challenges on solving the key problem and how we resolve these challenges.
>  The first challenge is that $\mathrm{K}$ is not available under federated settings. To this end, an estimate of $\mathrm{K}$, denoted as $\mathrm{\xi}$, is jointly constructed at users' devices based on random features [1] of local data samples. Based on $\mathrm{\xi}$, an estimate of $\mathrm{K}$, denoted as $\mathrm{Z}$ can be obtained at the cloud server. Afterwards, an approximate version of the top eigenpairs of $\mathrm{K}$ can be obtained from $\mathrm{Z}$ through singular value decomposition (SVD).
>
> The second challenge is how to improve the accuracy of approximation. To this end, $\mathrm{Z}$ is iteratively updated by DSPGD.
>   More specifically, in the $t$-th iteration, an estimate $\mathrm{\xi}\_{t}$ is constructed at users' devices, and then the estimate $\mathrm{Z}\_{t}$ at the cloud server is updated to $\mathrm{Z}\_{t+1}$ by applying DLA to a weighted sum of $\mathrm{Z}\_{t}$ and $\mathrm{\xi}\_{t}$.
>
>  In Section 4.2, we first point out that the process of obtaining an updated $\mathrm{Z}_{t}$ results in high communication cost, because DLA is applied to matrices with the number of rows/columns equal to the number of data samples. To this end, we design a communication efficient mechanism (CEM) so that DLA is applied to a new type of matrices with reduced dimensions. We then present the design of the new type of matrices and the procedure of obtaining the top eigenpairs of $\mathrm{K}$ in CEM.
>
>   In addition, to improve the clarity of the algorithm, we have moved the pseudo code of the algorithm back to the end of Section 4.2.
>
> 2. The paper is written in a convoluted manner. This is the main limitation, at some point, we are talking about $k$-means, SVS, DLA, DSPGD, EVD, SPGD, a bunch of other methods that are coupled together towards the main approach.
>
> We have revised the introduction and also Section 4.1 to clarify the relationship between our method with the bunch of other methods. For the detailed revision of the introduction, please refer to the response to comment 7.
>
> we first point out that the key problem for designing FK $k$-means is to obtain the top eigenpair of the kernel matrix $\mathrm{K}$ in a distributed manner. To solve this problem, we design a distributed stochastic proximal gradient descent (DSPGD) algorithm. We then present two challenges on solving the key problem and how we resolve these challenges.
>
>   The first challenge is that $\mathrm{K}$ is not available under federated settings. To this end, an estimate of $\mathrm{K}$, denoted as $\mathrm{\xi}$, is jointly constructed at users' devices based on random features [1] of local data samples. Based on $\mathrm{\xi}$, an estimate of $\mathrm{K}$, denoted as $\mathrm{Z}$ can be obtained at the cloud server. Afterwards, an approximate version of the top eigenpairs of $\mathrm{K}$ can be obtained from $\mathrm{Z}$ through singular value decomposition (SVD).
>
>   The second challenge is how to improve the accuracy of approximation. To this end, $\mathrm{Z}$ is iteratively updated by DSPGD.
>   More specifically, in the $t$-th iteration, an estimate $\mathrm{\xi}\_{t}$ is constructed at users' devices, and then the estimate $\mathrm{Z}\_{t}$ at the cloud server is updated to $\mathrm{Z}\_{t+1}$ by applying DLA to a weighted sum of $\mathrm{Z}\_{t}$ and $\mathrm{\xi}\_{t}$.
>
>   After the revision, the relationship between FK $k$-means and other methods is clear, and it is easier to see the novelty and contribution of our work from the revised algorithm description.

---

> > ### Author Response · Authors · 2020-11-24
> > **Response to Q3~Q8**
> >
> > 3.  Problem 1 seems to be an integer programming problem, thus with very high computation complexity. It is not clear how this is solved.
> >
> > Problem~1 is an NP-hard problem [4]. Thus, an approximate solution is required. One efficient method to determine the approximate solution is as follows [5]. The kernel matrix $\mathrm{K}$ is first decomposed as $\mathrm{K}={\mathrm{U}}{\mathrm{\Lambda}}{\mathrm{U}}^\top$ via SVD, and then a linear $k$-means algorithm is applied to the rows of the matrix ${\mathrm{H}}={\mathrm{U}}{\mathrm{\Lambda}}^{\frac{1}{2}}$ to obtain the approximate solution.
> >
> > 4. In the abstract please let me know what are those two levels of privacy you are talking about.
> >
> > The FK $k$-means provides two levels of privacy preservation: 1) users’ local data are not exposed to the cloud server; 2) the cloud server cannot recover users’ local data from the local computational results via matrix operations.
> >
> > 5. In the abstract, what does it mean that the clustering loss of the distributed method approaches the centralized one, please elaborate.
> >
> > In this statement we want to express that the clustering result of FK $k$-means approaches that of the standard kernel $k$-means, with an approximate ratio $(1+\varepsilon)$. The original statement ``the clustering loss of the distributed method approaches the centralized one'' is indeed confusing, and we have revised it in the abstract.
> >
> > 6. Why developing a federated learning algorithm is a promising approach? Please elaborate.
> >
> > Compared with the standard kernel $k$-means, the federated kernel $k$-means has two advantages: 1) users' raw data are not uploaded to the cloud server, which provides a basic level of privacy preservation; 2) it is usually more communication efficient to upload the local computational results than to upload the raw data to the cloud server. Thus, a federated kernel $k$-means algorithm is a promising approach.
> >
> > 7.  The second part of the intro turns into a detail technical analysis of the algorithm components, and so far we haven’t seen the algorithm so it all remains a technical abstract discussion that takes away the main messages.
> >
> > We have removed the detail technical analysis of the algorithm components from the second part of the introduction. Besides, we have revised the second part of the introduction as follows.
> >
> >   For the first challenging issue, we point out the key problem for designing FK $k$-means is to obtain the top eigenpair of the kernel matrix $\mathrm{K}$ in a distributed manner. To solve this problem, we design a distributed stochastic proximal gradient descent (DSPGD) algorithm. We then present two challenges on solving the key problem and how we resolve these challenges.
> >
> > The first challenge is that $\mathrm{K}$ is not available under federated settings. To this end, an estimate of $\mathrm{K}$, denoted as $\mathrm{\xi}$, is jointly constructed at users' devices based on random features [1] of local data samples. Based on $\mathrm{\xi}$, an estimate of $\mathrm{K}$, denoted as $\mathrm{Z}$ can be obtained at the cloud server. Afterwards, an approximate version of the top eigenpairs of $\mathrm{K}$ can be obtained from $\mathrm{Z}$ through singular value decomposition (SVD).
> >
> >   For the second challenging issue, we also point out how to improve the accuracy of approximation. To this end, $\mathrm{Z}$ is iteratively updated by DSPGD.
> >   More specifically, in the $t$-th iteration, an estimate $\mathrm{\xi}\_{t}$ is constructed at users' devices, and then the estimate $\mathrm{Z}\_{t}$ at the cloud server is updated to $\mathrm{Z}\_{t+1}$ by applying DLA to a weighted sum of $\mathrm{Z}\_{t}$ and $\mathrm{\xi}\_{t}$.
> >
> > 8. The algorithm is in the appendix, so the description and analysis is made on an item that has not been presented in the main text.
> >
> > We have moved the pseudo code of the algorithm back to Section 4.2.

---

> > > ### Author Response · Authors · 2020-11-24
> > > **Response to Q9**
> > >
> > > 9. The way the result is presented makes it look like the proposed method is a concatenation of other results, rather than the solution of a technical challenge in the problem.
> > >
> > > We have reorganized the presentation of the results to match the technical challenges.
> > >
> > >   We have added the following paragraph at the beginning of Section 5 to match the theoretical results with the challenges:
> > >   The convergence of DSPGD with CEM is analyzed in Section 5.1. The communication cost of CEM is analyzed in Section 5.2, which shows CEM is important for FK $k$-means to maintain the communication efficiency. It is then proved that the clustering results of FK $k$-means can approach that of the standard kernel $k$-means in Section 5.3. Besides, the privacy preservation provided by FK $k$-means is analyzed in Section 5.4.
> > >
> > >   We have also added the following paragraph in Section~6.2 to match the numerical results with the challenges:
> > >   The experimental results are presented in three aspects. First, the convergence results of DSPGD is shown in Figure 1 to verify its convergence rate. Second, the average communication cost per iteration of the two versions of DSPGD is provided in Figure 2 to show that CEM highly reduces the communication cost of DSPGD. Third, in Figure 3, FK $k$-means is compared with the cloud-based kernel $k$-means schemes in terms of clustering quality to show FK $k$-means can achieve the comparable clustering results as that of the cloud-based schemes; FK $k$-means is also compared with the existing distributed kernel $k$-means schemes under federated settings to show the higher communication efficiency of FK $k$-means.

---

### Author Response · Authors · 2020-11-25
**Summary of the revision of our paper**

We thank all the reviewers for their constructive suggestions on our paper.

We have resolved all the issues mentioned in the reviewers' comments. In addition, we have revised our paper as follows (marked by the red words in the revised version of our paper).

1.We have added the missing information to the abstract and revised the confusing statements in the abstract.

2.We have revised the description of FK $k$-means in the introduction by pointing out the key problems of designing FK $k$-means and how we solve these problems step by step.

3.We have added the comparison between our method with stochastic PCA and scalable kernel $k$-means in Section 2.

4.We have reorganized the description of FK $k$-means in Section 4.1 and Section 4.2, and have moved the pseudo code back to Section 4.2.

5.We have revised the main results of Theorem 1.

6.We have reorganized the presentation of theoretical results and experimental results to match the challenges on designing FK $k$-means.

---

### Decision · Program_Chairs · 2021-01-07
**Final Decision**

**Decision:**

Reject

**Comment:**

This paper presents a approach to the distributed kernel k-means problem using a combination of random features to efficiently approximate the kernel matrix, a distributed stochastic proximal gradient algorithm which calls a distributed lanczos algorithm as a primitive to find a low-rank approximation to the kernel matrix, and additional compression to reduce the cost of the communications.

The algorithm is a novel combination of prior ideas, and empirically works well. However, the claimed theoretical convergence rate is not convincing: e.g., the convergence rate depends on the Frobenius norm of the error in approximating the kernel with random feature maps, which is O(n^2) for a problem with n data samples. This implies that O(n^2) iterations must be used in the algorithm, which is already slower than a naive approach to kernel k-means.

This paper takes a promising approach to the problem, but as the potential contribution lies in combining prior ideas in order to obtain a provably guaranteed approximate solution to the distributed kernel k-means problem, and the proposed algorithm was not shown to satisfy this promise, the recommendation is to reject.